# Learning Mixture Models with Simultaneous Data Partitioning and Parameter Estimation

## Abstract

We study a new framework of learning mixture models via data partitioning called PRESTO, wherein we optimize a joint objective function on the model parameters and the partitioning, with each model tailored to perform well on its specific partition. In contrast to prior work, we do not assume any generative model for the data. We connect our framework to a number of past works in data partitioning, mixture models, and clustering, and show that PRESTO generalizes several loss functions including the k-means, Bregman clustering objective, the Gaussian mixture model objective, mixtures of support vector machines, and mixtures of linear regression. We convert our training problem to a joint parameter estimation cum a subset selection problem, subject to a matroid span constraint. This allows us to reduce our problem into a constrained set function minimization problem, where the underlying objective is monotone and approximately submodular. We then propose a new joint discrete-continuous optimization algorithm which achieves a bounded approximation guarantee for our problem. We show that PRESTO outperforms several alternative methods. Finally, we study PRESTO in the context of resource efficient deep learning, where we train smaller resource constrained models on each partition and show that it outperforms existing data partitioning and model pruning/knowledge distillation approaches, which in contrast to PRESTO, require large initial (teacher) models.

## 1 Introduction

In the problem space of learning mixture models, our goal is to fit a given set of models implicitly to different clusters of the dataset. Mixture models are ubiquitous approaches for prediction tasks on heterogeneous data (Dasgupta, 1999; Achlioptas & McSherry, 2005; Kalai et al., 2010; Belkin & Sinha, 2010a; Pace & Barry, 1997; Belkin & Sinha, 2010b; Sanjeev & Kannan, 2001; Hopkins & Li, 2018; Fu & Robles-Kelly, 2008), and find use in a plethora of applications, *e.g.*, finance, genomics (Dias et al., 2009; Liesenfeld, 2001; Pan et al., 2003), *etc*. Existing literature on mixture models predominately focuses on the design of estimation algorithms and the analysis of sample complexity for these problems (Faria & Soromenho, 2010; Städler et al., 2010; Kwon et al., 2019; Yi et al., 2014), and analyzes them theoretically for specific and simple models such as Gaussians, linear regression, and SVMs. Additionally, erstwhile approaches operate on *realizable* settings — they assume specific generative models for the cluster membership of the instances. Such an assumption can be restrictive, especially when the choice of the underlying generative model differs significantly from the hidden data generative mechanism. Very recently, Pal et al. (2022) consider a linear regression problem in a *non-realizable* setting, where they do not assume any underlying generative model for the data. However, their algorithm and analysis is tailored towards the linear regression task.

### 1.1 Present work

Responding to the above limitations, we design PRESTO, a novel data partitioning based framework for learning mixture models. In contrast to prior work, PRESTO is designed for generic deep learning problems including classification using nonlinear architectures, rather than only linear models (linear regression or SVMs). Moreover, we do not assume any generative model for the data. We summarize our contributions as follows.

**Novel framework for training mixture models.** At the outset, we aim to simultaneously partition the instances into different subsets and build a mixture of models across these subsets. Here, each model is tailored to perform well on a specific portion of the instance space. Formally, given a set of instances and the architectures of $K$ models, we partition the instances into $K$ disjoint subsets and train a family of $K$ component models on these subsets, wherein, each model is assigned to one subset, implictly by the algorithm. Then, we seek to minimize the sum of losses yielded by the models on the respective subsets, jointly with respect to the model parameters and the candidate partitions of the underlying instance space.

Note that our proposed optimization method aims to attach each instance to one of the $K$ models on which it incurs the least possible error. Such an approach requires that the loss function helps guide the choice of the model for an instance, thus rendering it incompatible for use at inference time. We build an additional classifier to tackle this problem; given an instance, the classifier takes the confidence from each of the $K$ models as input and predicts the model to be assigned to it.

**Design of approximation algorithm.** Our training problem involves both continuous and combinatorial optimization variables. Due to the underlying combinatorial structure, the problem is NP-hard even when all the models are convex. To solve this problem, we first reduce our training problem to a parameter estimation problem in conjunction with a subset selection task, subject to a matroid span constraint (Iyer et al., 2014). Then, we further transform it into a constrained set function minimization problem and show that the underlying objective is a monotone and $\alpha$-submodular function (El Halabi & Jegelka, 2020; Gatmiry & Gomez-Rodriguez, 2018) and has a bounded curvature. Finally, we design PRESTO, an approximation algorithm that solves our training problem, by building upon the majorization-minimization algorithms proposed in (Iyer & Bilmes, 2015; Iyer et al., 2013a; Durga et al., 2021). We provide the approximation bounds of PRESTO, even when the learning algorithm provides an imperfect estimate of the trained model. Moreover, it can be used to minimize any $\alpha$-submodular function subject to a matroid span constraint and therefore, is of independent interest.

**Application to resource-constrained settings.** With the advent of deep learning, the complexity of machine learning (ML) models has grown rapidly in the last few years (Liu et al., 2020; Arora et al., 2018; Dar et al., 2021; Bubeck & Sellke, 2021; Devlin et al., 2018; Liu et al., 2019; Brown et al., 2020). The functioning of these models is strongly contingent on the availability of high performance computing infrastructures, *e.g.*, GPUs, large RAM, multicore processors, *etc*.

The key rationale behind the use of an expensive neural model is to capture the complex nonlinear relationship between the features and the labels across the entire dataset. Our data partitioning framework provides a new paradigm to achieve the same goal, while enabling multiple lightweight models to run on a low resource device. Specifically, we partition a dataset into smaller subslices and train multiple small models on each subslice— since each subslice is intuitively a smaller and simpler data subset, we can train a much simpler model on the subslice thereby significantly reducing the memory requirement. In contrast to approaches such as pruning (Wang et al., 2020a; Lee et al., 2019; Lin et al., 2020; Wang et al., 2020b; Jiang et al., 2019; Li et al., 2020; Lin et al., 2017) and knowledge distillation (Hinton et al., 2015; Son et al., 2021), we do not need teacher models (high compute models) with the additional benefit that we can also train these models on resource constrained devices.

**Empirical evaluations.** Our experiments reveal several insights, summarized as follows. (1) PRESTO yields significant accuracy boost over several baselines. (2) PRESTO is able to trade-off accuracy and memory consumed during training more effectively than several competitors, *e.g.*, pruning and knowledge distillation approaches. At the benefit of significantly lower memory usage, the performance of our framework is comparable to existing pruning and knowledge distillation approaches and much better than existing partitioning approaches and mixture models.

## 1.2 RELATED WORK

**Mixture Models and Clustering:.** Mixture Models (Dempster et al., 1977; Jordan & Jacobs, 1994) and k-means Clustering (MacQueen, 1967; Lloyd, 1982) are two classical ML approaches, and have seen significant research investment over the years. Furthermore, the two problems are closely connected and the algorithms for both, *i.e.*, the k-means algorithm and the Expectation Maximization algorithm for mixture models are closely related – the EM algorithm is often called soft-clustering,

wherein one assigns probabilities to each cluster. Mixture models have been studied for a number of problems including Gaussian Mixture Models (Xu & Jordan, 1996), Mixtures of SVMs (Collobert et al., 2001; Fu & Robles-Kelly, 2008), and linear regression (Faria & Soromenho, 2010; Städler et al., 2010; Kwon et al., 2019; Pal et al., 2022). As we will show in this work, our loss based data partitioning objective generalizes the objectives of several tasks, including k-means clustering (Lloyd, 1982), Clustering with Bregman divergences (Banerjee et al., 2005), and mixture models with SVMs and linear regression (Collobert et al., 2001; Pal et al., 2022).

Following initial analysis of the EM for Gaussian Models (Jordan & Jacobs, 1994; Xu & Jordan, 1996), mixture models have also been studied for SVMs (Collobert et al., 2001), wherein the authors extend the mixture-model formulation to take the SVM loss instead of the Gaussian distribution. A lot of recent work has studied mixtures of linear regression models. Most prior work involving mixture models pertaining to regression is in the *realizable setting*, with the exception of (Pal et al., 2022). Mixture model papers aim at using the expectation-maximisation (EM) algorithm for parameter estimation. Balakrishnan et al. (2017) study the EM algorithm which is initialised with close estimates to ground truth, concluding that it is able to give reasonable fits to the data. When the sample size is finite, (Balakrishnan et al., 2017) show convergence within $l_2$ norm of the true parameters. Yi et al. (2014) provide yet another initialisation procedure which uses eigenspace analysis and works for two bimodal regressions.These works assume that the underlying Gaussian distributions have the same shared covariance. Li & Liang (2018) lift this assumption, proving near-optimal complexity. Pal et al. (2022) approach the problem of linear regression under the *non-realizable setting* by defining *prediction* and *loss* in the context of mixtures and formulating a novel version of the alternating maximisation (AM) algorithm.

**Resource-constrained learning.** In the pursuit of better performance, most state of the art deep learning models are often over-parameterized. This makes their deployment in the resource constrained devices nearly impossible. To mitigate the problems, several handcrafted architectures such as SqueezeNets (Iandola et al., 2016; Gholami et al., 2018), MobileNets (Howard et al., 2017; Sandler et al., 2018) and ShuffleNets (Zhang et al., 2018; Ma et al., 2018) were designed to work in mobile devices. Recently, EfficientNet (Tan & Le, 2019) was proposed, that employs neural architecture search. However, these models are designed to work on the entire training set, and leave a heavy memory footprint.

**Simultaneous model training and subset selection.** Data subset selection approaches are predominately static, and most often, do not take into account the model's current state (Wei et al., 2014a;b; Kirchhoff & Bilmes, 2014; Kaushal et al., 2019; Liu et al., 2015; Bairi et al., 2015; Lucic et al., 2017; Campbell & Broderick, 2018; Boutsidis et al., 2013). Some recent approaches attempt to solve the problem of subset selection by performing joint training and subset selection (Mirzasoleiman et al., 2020a;b; Killamsetty et al., 2021b;a; Durga et al., 2021). On the other hand, some other approaches (Mirzasoleiman et al., 2020a; Killamsetty et al., 2021a) select subsets that approximate the full gradient. Further, some of these approaches (Killamsetty et al., 2021b;a) demonstrate improvement in the robustness of the trained model by selecting subsets using auxiliary or validation set. Durga et al. (2021) train the model in such a way that the validation set error is controlled. All the aforementioned techniques try to determine (for purposes of training), a single subset of the training set.In this paper we propose PRESTO as a method to simultaneously select multiple subsets and also learn mixture of models (with a model corresponding to each subset), in order to improve robustness of the trained model, even in resource-constrained settings.

## 2 PROBLEM FORMULATION

In this section, we first present the notations and then formally state our problem. Finally, we instantiate our problem in different clustering and mixture modeling scenarios.

### 2.1 NOTATIONS

We have $N$ training instances $\{(\boldsymbol{x}_i, y_i) \,|\, 1 \leq i \leq N\}$, where $\boldsymbol{x}_i \in \mathcal{X}$ is the feature vector and $y_i \in \mathcal{Y}$ is the label of the $i^{\text{th}}$ instance. In our work, we set $\mathcal{X} = \mathbb{R}^d$ and treat $\mathcal{Y}$ to be discrete[1]. Given $K$, we denote $h_{\boldsymbol{\theta}_1}, .., h_{\boldsymbol{\theta}_K}$ as the component models that are going to be used for the $K$ subsets resulting from a partition of $\mathcal{X}$. Here $\boldsymbol{\theta}_k$ is the trainable parameter vector of $h_{\boldsymbol{\theta}_k}$. These parameters are not

---

[1]For brevity, we present our analysis for the classification setup. However, our framework is also applicable, as-it-is, to the regression setup.

shared across different models, *i.e.*, there is no overlap between $\boldsymbol{\theta}_k$ and $\boldsymbol{\theta}_{k'}$ for $k \neq k'$. In fact, the models $h_{\boldsymbol{\theta}_k}$ and $h_{\boldsymbol{\theta}_{k'}}$ can even have different architectures. Moreover, the representation of the output $h_{\boldsymbol{\theta}_k}$ can vary across different settings. For example, given $\boldsymbol{x} \in \mathcal{X}$, $\text{sign}(h_{\boldsymbol{\theta}_k}(\boldsymbol{x}))$ is a predictor of $y$ for support vector machines with $\mathcal{Y} \in \{\pm 1\}$ whereas, for multiclass classification, $h_{\boldsymbol{\theta}_k}(\boldsymbol{x})$ provides a distribution over $\mathcal{Y}$. To this end, we use $\ell(h_{\boldsymbol{\theta}_k}(\boldsymbol{x}), y)$ to indicate the underlying loss function for any instance $(\boldsymbol{x}, y)$. We define $[A] = \{1, \ldots, A\}$ for an integer $A$.

## 2.2 PROBLEM STATEMENT

**High level objective.** Our broad goal is to fit a mixture of models on a given dataset, without making any assumption concerning the generative process, the instances or features. Given a family of model architectures, our aim is to learn to partition the instance space $\mathcal{X}$ into a set of subsets, determine the appropriate model architecture to be assigned to each subset and to subsequently train the appropriate models on the respective subsets.

**Problem statement.** We are given the training instances $D$, the number of subsets $K$ resulting from partitioning $D$ and a set of model architectures $h_{\boldsymbol{\theta}_1}, \ldots, h_{\boldsymbol{\theta}_K}$. Our goal, then, is to partition the training set $D$ into $K$ subsets $S_1, ..S_K$ with $S_k \cap S_{k'} = \emptyset$ and $\cup_k S_k = D$ so that when the model $h_{\boldsymbol{\theta}_k}$ is trained on $S_k$ for $k \in [K]$, the total loss is minimized. To this end, we define the following regularized loss, *i.e.*,

$$F(\{(S_k, \boldsymbol{\theta}_k) \,|\, k \in [K]\}) = \sum_{k \in [K]} \sum_{i \in S_k} \left[ \ell \left( h_{\boldsymbol{\theta}_k}(\boldsymbol{x}_i), y_i \right) + \lambda ||\boldsymbol{\theta}_k||^2 \right]. \tag{1}$$

Then, we seek to solve the following constrained optimization problem:

$$\underset{\substack{\boldsymbol{\theta}_1, \ldots, \boldsymbol{\theta}_K, \\ S_1, \ldots S_K}}{\text{minimize}} \; F(\{(S_k, \boldsymbol{\theta}_k) \,|\, k \in [K]\}) \;\; \text{subject to,} \;\; S_k \cap S_{k'} = \emptyset, \; \forall k \in [K], \;\; \cup_{k \in [K]} S_k = D. \tag{2}$$

Here, $\lambda$ is the coefficient of the $L_2$ regularizer in Eq. (2). The constraint $S_k \cap S_{k'} = \emptyset$ ensures that that each example $i \in D$ belongs to exactly one subset and the constraint $\cup_{k \in [K]} S_k = D$ entails that the subsets $\{S_k \,|\, k \in [K]\}$ cover the entire dataset. The above optimization problem is a joint model parameter estimation and a partitioning problem. If we fix the partition $\{S_1, S_2, ..., S_K\}$, then the optimal parameter vector $\boldsymbol{\theta}_k$ depends only on $S_k$, the subset assigned to it. To this end, we denote the optimal value of $\boldsymbol{\theta}_k$ for the above partition as $\boldsymbol{\theta}_k^*(S_k)$ and transform the optimization (2) into the following equivalent problem:

$$\underset{S_1, S_2, \ldots, S_K}{\text{minimize}} \; F(\{S_k, \boldsymbol{\theta}_k^*(S_k) \,|\, k \in [K]\}) \;\;\; \text{subject to,} \; S_k \cap S_{k'} = \emptyset, \; \forall k \in [K], \cup_{k \in [K]} S_k = D. \tag{3}$$

Note that, computing $\boldsymbol{\theta}_k^*(S_k)$ has a polynomial time complexity for convex loss functions. However, even for such functions, minimizing $F$ as defined above is NP-hard.

*Test time prediction.* Since computation of the optimal partitioning requires us to solve the optimization (3), it cannot be used to assign an instance $\boldsymbol{x}$ to a model $h_{\boldsymbol{\theta}_k}$ during the test time. To get past this blocker, we train an additional multiclass classifier $\pi_\phi : \mathcal{X} \to [K]$, which is trained on $\{(\boldsymbol{x}_i, k)\}$ pairs where $i \in S_k$, so that, during test time, it can assign an unseen instance $\boldsymbol{x}$ to a model component $h_{\boldsymbol{\theta}_k}$ using $k = \pi_\phi(\boldsymbol{x})$.

## 2.3 INSTANTIATIONS IN CLUSTERING AND MIXTURE MODELING

In this section, we show how the formulations listed in the previous section (*e.g.*, Eqn (1)) have appeared in several applications ranging from clustering to mixture modeling.

**K-Means Clustering:** Since k-means clustering (Lloyd, 1982; MacQueen, 1967) is unsupervised, we do not have access to labels $y_i$. It then turns out that $\theta_k = \mu_k$, the cluster means, and $\ell(h_{\boldsymbol{\theta}_k}(\boldsymbol{x}_i)) = ||\boldsymbol{x}_i - \mu_k||^2$.

**Bregman Clustering:** K-means clustering is a special case of the more general clustering with Bregman divergences (Banerjee et al., 2005). Again, in this case, the parameters $\theta_k = \mu_k$, the cluster means and $\ell(h_{\boldsymbol{\theta}_k}(\boldsymbol{x}_i)) = B_\phi(\boldsymbol{x}_i, \mu_k)^2$.

**Mixture of SVMs:** The mixture of Support Vector Machines (Fu & Robles-Kelly, 2008) is a special case of Eq. (1) with $\ell(h_{\boldsymbol{\theta}_k}(\boldsymbol{x}_i), y_i) = \max(0, 1 - h_{\boldsymbol{\theta}_k}(\boldsymbol{x}_i) y_i)$.

---

[2]The Bregman divergence is defined as: $B_\phi(x, y) = \phi(x) - \phi(y) - \langle \nabla \phi(y), x - y \rangle$.

**Mixture of Linear Regression:** The mixture of linear regression models (Pal et al., 2022) is a special case of Eqn. (1) with $\ell(h_{\boldsymbol{\theta}_k}(\boldsymbol{x}_i), y_i) = ||h_{\boldsymbol{\theta}_k}(\boldsymbol{x}_i) - y_i||^2$.

## 2.4 APPLICATION IN THE RESOURCE-CONSTRAINED SETUP

In general, the relationship between an instance $\boldsymbol{x}$ and the label $y$ can be arbitrarily nonlinear. A complex deep neural network captures such relationship by looking into the entire dataset. However, such networks require high performance computing infrastructure for training and inference. The training problem (3) can be used to build $K$ lightweight models $h_{\boldsymbol{\theta}_1}, \ldots, h_{\boldsymbol{\theta}_K}$, each of which is localized to a specific regime of the instance space $\mathcal{X}$. Thus, a model is required to capture the nonlinearity only from the region assigned to it and not the entire set $\mathcal{X}$. As a result, it can be lightweight and used with limited resources.

Consider a large model with number of parameters equal to the total number of parameters collectively across the $K$ models, *i.e.*, $\sum_{k=1}^{K} \dim(\boldsymbol{\theta}_k)$. Such a model has to be loaded entirely into a GPU RAM for training or inference, and thus requires a larger GPU RAM. In contrast, our approach requires us to load at a time, only one model component $h_{\boldsymbol{\theta}_k}$ and the corresponding subset $S_k$ during both training and test. This is instead of having to load all the $K$ model components and the entire dataset. While this can increase both training and inference time, it can substantially reduce the memory consumption by $1/K$ times in comparison to a large model having similar expressiveness.

## 3 PRESTO: PROPOSED FRAMEWORK TO SOLVE THE TRAINING PROBLEM (3)

In this section, we first show that the optimization problem (3) is equivalent to minimizing a monotone set function subject to matroid span constraint (Iyer et al., 2014; Schrijver et al., 2003). Subsequently, we show that this set function is $\alpha$-submodular and admits a bounded curvature. Finally, we use these results to design PRESTO, an approximation algorithm to solve the underlying constrained set function optimization problem. We next present these analyses, beginning with the necessary definitions about monotonicity, $\alpha$-submodularity and different matroid related properties.

**Definition 1. (1) Monotonicity, $\alpha$-submodularity and generalized curvature:** Given a ground set $\mathbb{V}$, let $G : 2^{\mathbb{V}} \to \mathbb{R}$ be a set function whose marginal gain is denoted by $G(e \,|\, \mathbb{S}) = G(\mathbb{S} \cup \{e\}) - G(\mathbb{S})$. The function $G$ is monotone non-decreasing if $G(e \,|\, \mathbb{S}) \geq 0$ for all $\mathbb{S} \subset \mathbb{V}$ and $e \in \mathbb{V}\backslash\mathbb{S}$. $G$ is $\alpha$-submodular with $\alpha \in (0, 1]$ if $G(e \,|\, \mathbb{S}) \geq \alpha G(e \,|\, \mathbb{T})$ for all $\mathbb{S} \subseteq \mathbb{T}$ and $e \in \mathbb{V}\backslash\mathbb{T}$ (Hashemi et al., 2019; El Halabi & Jegelka, 2020). The generalized curvature of $G(\mathbb{S})$ is defined as $\kappa_G(\mathbb{S}) = 1 - \min_{e \in \mathbb{S}} G(e \,|\, \mathbb{S}\backslash e)/G(e \,|\, \emptyset)$ (Iyer et al., 2013b; Zhang & Vorobeychik, 2016). **(2) Base, rank and span of a matroid:** Consider a matroid $\mathcal{M} = (\mathbb{V}, \mathcal{I})$ where $\mathcal{I}$ is the set of independent sets (Refer to Appendix A.1 for more details). A base of $\mathcal{M}$ is a maximal independent set. The rank function $r_{\mathcal{M}} : 2^{\mathbb{V}} \to \mathbb{N}$ is defined as: $r_{\mathcal{M}}(\mathbb{S}) = \max_{I \in \mathcal{I}:I \subset \mathbb{S}} |I|$. A set $\mathbb{S}$ is a spanning set if it is the superset of a base or equivalently, $r_{\mathcal{M}}(\mathbb{S}) = r_{\mathcal{M}}(\mathbb{V})$ (Schrijver et al., 2003; Iyer et al., 2014; Edmonds, 2003).

### 3.1 REPRESENTATION OF (3) AS A MATROID SPAN CONSTRAINED SUBSET SELECTION TASK

**Transforming partitions into 2D configurations.** Given the training instances $D$ and the size of partition $K$, we first define the ground set $\mathbb{V}$ in the space of Cartesian products of $D$ and $[K]$, *i.e.*, $\mathbb{V} = D \times [K]$. Thus $\mathbb{V}$ consists of all pairs $\{(i,k) \,|\, i \in D, k \in [K]\}$ which enumerates all possible assignments between the instances and model components. Moreover, we define $\mathbb{V}_{i\star} = \{(i,k) \,|\, k \in [K]\}$ and $\mathbb{V}_{\star k} = \{(i,k) \,|\, i \in D\}$. Here, $\mathbb{V}_{i\star} = \{i\} \times [K]$ enumerates all possible assignments of the $i^{\text{th}}$ instance and $\mathbb{V}_{\star k} = D \times \{k\}$ enumerates all possible configurations specifically wherein $S_k$ is assigned to an instance.

**Reformulating optimization (3) in constrained subset selection.** Having defined the ground set $\mathbb{V}$ and the partitions in 2D configuration space as above, we define $\mathbb{S} = \{(i,k) \,|\, i \in S_k, k \in [K]\}$ that encodes the set of assignments in space of $\mathbb{V}$ induced by the underlying partition. Then, the set $\widehat{\mathbb{S}}_k = \{(i,k) \,|\, i \in S_k\}$ specifies the set of instances attached to the subset $k$ and elucidates the subset containing $i$. It can be observed that $\widehat{\mathbb{S}}_k = \mathbb{S} \cap \mathbb{V}_{\star k}$. Since every instance is assigned exactly to one subset $S_k \in \{S_1, \ldots, S_K\}$, we have that $|\mathbb{S} \cap \mathbb{V}_{i\star}| = 1$. To this end, we introduce the following set function, which is the sum of the fitted loss functions defined in Eq. (1), trained over individual

subsets.

$$G(\mathbb{S}) = \sum_{k \in [K]} \sum_{(i, \bullet) \in \mathbb{S} \cap \mathbb{V}_{\star k}} \left[ \ell \left( h_{\boldsymbol{\theta}_k^*(\mathbb{S} \cap \mathbb{V}_{\star k})}(\boldsymbol{x}_i), y_i \right) + \lambda ||\boldsymbol{\theta}_k||^2 \right]. \tag{4}$$

We rewrite our optimization problem (3) as follows:

$$\underset{\mathbb{S} \subset \mathbb{V}}{\text{minimize}} \ G(\mathbb{S}) \text{ subject to, } |\mathbb{S} \cap \mathbb{V}_{i\star}| = 1 \ \forall \ i \in D. \tag{5}$$

**Theoretical characterization of the objective and constraints in Eq. (5).** Here, we show that our objective admits monotonicity, $\alpha$-submodularity and bounded curvature in a wide variety of scenarios (Proven in Appendix A.2)

**Theorem 2.** *Given the set function $G(\mathbb{S})$ defined in Eq. (4) and the individual regularized loss functions $\ell$ introduced in Eq. (1), we define: $\varepsilon_{\min} = \max_{k,i} Eigen_{\min}[\nabla^2_{\boldsymbol{\theta}_k} \ell(h_{\boldsymbol{\theta}_k}(\boldsymbol{x}_i), y_i)]$, $\underline{\ell_{\min}} = \min_{i \in D \boldsymbol{\theta}_k}[\ell(h_{\boldsymbol{\theta}_k}(\boldsymbol{x}_i), y_i) + \lambda ||\boldsymbol{\theta}_k||^2$ and $\overline{\ell_{\min}} = \max_{i \in D} \min_{\boldsymbol{\theta}_k}[\ell(h_{\boldsymbol{\theta}_k}(\boldsymbol{x}_i), y_i) + \lambda ||\boldsymbol{\theta}_k||^2]$. Then, we have the following results:*

*(1) Monotonicity: The function $G(\mathbb{S})$ defined in Eq. (4) is monotone non-decreasing in $\mathbb{S}$.*

*(2) $\alpha$-submodularity: If $\ell(h_{\boldsymbol{\theta}_k}(\boldsymbol{x}), y)$ is L-Lipschitz for all $k \in \{1, .., K\}$ and the regularizing coefficient $\lambda$ satisfies: $\lambda > -\varepsilon_{\min}$, function $G(\mathbb{S})$ is $\alpha$-submodular with*

$$\alpha \geq \alpha_G = \frac{\underline{\ell_{\min}}}{\underline{\ell_{\min}} + \frac{2L^2}{\lambda + 0.5\varepsilon_{\min}} + \frac{\lambda L^2}{(\lambda + 0.5\varepsilon_{\min})^2}} \tag{6}$$

*(3) Generalized curvature: The generalized curvature $\kappa_G(\mathbb{S})$ for any set $\mathbb{S}$ is given by: $\kappa_G(S) \leq \kappa_G^* = 1 - \underline{\ell_{\min}}/\overline{\ell_{\min}}$.*

Solving the optimization (5) is difficult due to the equality constraint. However, as suggested by Theorem 2 (1), $G(\mathbb{S})$ is monotone in $\mathbb{S}$. Hence, even if we relax the equality constraints $|\mathbb{S} \cap \mathbb{V}_{i\star}| = 1$ to the inequality constraint $|\mathbb{S} \cap \mathbb{V}_{i\star}| \geq 1$, they achieve the equality at the optimal solution $\mathbb{S}^*$. Thus, the optimization (5) becomes

$$\underset{\mathbb{S} \subset \mathbb{V}}{\text{minimize}} \ G(\mathbb{S}) \text{ subject to, } |\mathbb{S} \cap \mathbb{V}_{i\star}| \geq 1 \ \forall \ i \in D. \tag{7}$$

As we formally state in the following proposition, the above constraint (set) can be seen as a matroid span constraint for a partition matroid. This would allow us to design an approximation algorithm to solve this problem.

**Proposition 3.** *Suppose that set $\mathbb{S}$ satisfies $|\mathbb{S} \cap \mathbb{V}_{i\star}| \geq 1$ for all $i \in D$. Then $\mathbb{S}$ is a spanning set of the partition matroid $\mathcal{M} = (\mathbb{V}, \mathcal{I})$ where $\mathcal{I} = \{I \,|\, |I \cap \mathbb{V}_{i\star}| \leq 1\}$. Moreover, if $\mathbb{S}$ satisfies $|\mathbb{S} \cap \mathbb{V}_{i\star}| = 1$ for all $i \in D$, then $\mathbb{S}$ is a base of $\mathcal{M}$.*

The above proposition suggests that the optimal solution $\mathbb{S}^*$ of the optimization (7) is a base of the partition matroid $\mathcal{M} = (\mathbb{V}, \mathcal{I})$ with $\mathcal{I} = \{I \,|\, |I \cap \mathbb{V}_{i\star}| \leq 1\}$.

## 3.2 PRESTO: AN APPROXIMATION ALGORITHM TO SOLVE THE OPTIMIZATION PROBLEM (7)

In this section, we present our approximation algorithm PRESTO for minimizing the optimization problem (5), which is built upon the algorithm proposed by Durga et al. (2021). Their algorithm aims to approximately minimize an $\alpha$-submodular function, whereas we extend their algorithm for minimizing an $\alpha$-submodular function with matroid span constraint and derive an approximation guarantee for it. Specifically, we employ the Majorization-Minimization approach (Iyer et al., 2013a;b) for minimizing the optimization problem (5). Toward that goal, we first develop the necessary ingredients as follows.

**Computation of modular upper bound.** First, we present a modular upper bound for $G(\mathbb{S})$ (Iyer et al., 2013a). Given a fixed set $\widehat{\mathbb{S}}$ and the set function $G$ which is $\alpha$-submodular and monotone, let the modular function $m_{\widehat{\mathbb{S}}}^G[\mathbb{S}]$ be defined as follows:

$$m_{\widehat{\mathbb{S}}}^G[\mathbb{S}] = G(\widehat{\mathbb{S}}) - \sum_{(i,k) \in \widehat{\mathbb{S}}} \alpha_G G((i,k) \,|\, \widehat{\mathbb{S}} \setminus \{(i,k)\}) + \sum_{(i,k) \in \widehat{\mathbb{S}} \cap \mathbb{S}} \alpha_G G((i,k) \,|\, \widehat{\mathbb{S}} \setminus \{(i,k)\}) + \sum_{(i,k) \in \mathbb{S} \setminus \widehat{\mathbb{S}}} \frac{G((i,k) \,|\, \emptyset)}{\alpha_G}. \tag{8}$$

If $G$ is $\alpha$-submodular and monotone, it holds that $G(\mathbb{S}) \leq m_{\widehat{\mathbb{S}}}^G[\mathbb{S}]$ for all $\mathbb{S} \subseteq \mathcal{D}$ (Durga et al., 2021). Note that when $G$ is submodular, *i.e.*, when $\alpha = 1$, the expression $m_{\widehat{\mathbb{S}}}^G[\mathbb{S}]$ gives us the existing modular upper bounds for submodular functions (Nemhauser et al., 1978; Iyer et al., 2013a; Iyer & Bilmes, 2012).

**Outline of PRESTO (Alg. 1).** The goal of Algorithm 1 is to iteratively minimize $m_{\widehat{\mathbb{S}}}^G[\mathbb{S}]$ with respect to $\mathbb{S}$ in a majorization-minimization manner (Iyer et al., 2013a;b; Durga et al., 2021). Having computed $\mathbb{S}$ that minimizes $m_{\widehat{\mathbb{S}}}^G[\mathbb{S}]$ in the iteration $r - 1$, we set $\widehat{\mathbb{S}} = \mathbb{S}$ for the iteration $r$ and minimize $m_{\widehat{\mathbb{S}}}^G[\mathbb{S}]$ with respect to $\mathbb{S}$.

We start with $\widehat{\mathbb{S}} = \emptyset$ (the empty partition) in line 2. Since we minimize $m_{\widehat{\mathbb{S}}}^G[\mathbb{S}]$ with respect to $\mathbb{S}$, the key components for calculation are the last two terms of the RHS in Eq. (8), *i.e.*, $G(\{(i, k)\})/\alpha_G$ and $G((i, k) \mid \widehat{\mathbb{S}} \setminus \{(i, k\})$. We pre-compute the former in lines 3–8 for all pairs $(i, k)$. The complexity of this operation is $O(|D|K)$. We can compute $G((i, k) \mid \widehat{\mathbb{S}} \setminus \{(i, k\})$ by finding the corresponding partitions $\{\widehat{S}_1, \cdots, \widehat{S}_K\}$ and for each $k$, compute $G(\widehat{S}_k)$ and $G(\widehat{S}_k \setminus (i, k)), (i, k) \in \widehat{\mathbb{S}}_k$ (lines 13–16). Finally, once we compute the modular upper bound $m_{\widehat{\mathbb{S}}}^G[\mathbb{S}]$, we minimize it

---

**Algorithm 1** The PRESTO Algorithm

**Require:** Training data $D$, $\alpha_G$, $K$ model architectures, `Iterations`.
1: **Output:** The learned parameters $\{\boldsymbol{\theta}_k \mid k \in [K]\}$, the partitioning of $D$: $D = \cup_{k \in [K]} \widehat{S}_k$
2: $\widehat{S}_1 \leftarrow \emptyset, .., \widehat{S}_K \leftarrow \emptyset$
3: **for** $k \in [K]$ **do**
4:     $\widehat{\boldsymbol{\theta}}_k \leftarrow \text{INITPARAMS}( )$
5:     **for all** $i \in D$ **do**
6:         $G(\{(i, k)\}) \leftarrow \text{Train}(\{i\}; \widehat{\boldsymbol{\theta}}_k)$
7:     **end for**
8: **end for**

9: **for** $r \in [\text{Iterations}]$ **do**
10:     **for** $k \in [K]$ **do**
11:         $\widehat{\mathbb{S}}_k \leftarrow \left\{ (i, k) \mid i \in \widehat{S}_k \right\}$
12:         $\widehat{\boldsymbol{\theta}}_k, G(\widehat{\mathbb{S}}_k) \leftarrow \text{Train}(\widehat{S}_k; \boldsymbol{\theta}_k)$
13:         **for** $i \in \widehat{S}_k$ **do**
14:             $G(\widehat{\mathbb{S}}_k \setminus (i, k)) \leftarrow \text{Train}(\widehat{S}_k \setminus i; \boldsymbol{\theta}_k)$
15:             $\boldsymbol{M}[i][k] \leftarrow \alpha_G[G(\widehat{\mathbb{S}}_k) - G(\widehat{\mathbb{S}}_k \setminus \{(i, k)\})]$
16:         **end for**
17:         for all $i \notin \widehat{S}_k$, set $\boldsymbol{M}[i][k] = \dfrac{G(\{(i, k)\})}{\alpha_G}$
18:     **end for**
19:     $(i^*, k^*) \leftarrow \arg\min_{i,k}(\boldsymbol{M}[i][k])$
20:     $\widehat{S}_{k^*} \leftarrow \widehat{S}_{k^*} \cup i^*$
21:     For all $k \neq k^* : S_k \leftarrow S_k \setminus \{i^*\}$
22:     $\widehat{\mathbb{S}} \leftarrow \left\{ (i, k) \mid i \in \widehat{S}_k, k \in [K] \right\}$
23: **end for**
24: Return $\widehat{\boldsymbol{\theta}}_1, ..., \widehat{\boldsymbol{\theta}}_K, \widehat{\mathbb{S}}$

---

subject to the matroid span constraint– thus, we get a partition. Since $G$ is a modular function, we are guaranteed to find a set which is in the base of the partition matroid. Finally, note that to evaluate $G$, we need to train the algorithm on the specific datapoints and partition, as encapsulated in the Train( ) routine in Algorithm 1. Here, $\text{Train}(S, \theta_k)$ trains the model for partition $k$ on the subset $\widehat{S}_k$ for a few iterations and, returns the estimated parameters $\widehat{\boldsymbol{\theta}}_k$ and the objective $G(\widehat{\mathbb{S}}_k)$.

**Approximation guarantee.** The following theorem shows that if $G$ is $\alpha$-submodular with $\alpha > \alpha_G$, and curvature $\kappa > \kappa_G$, Algorithm 1 enjoys an approximation guarantee (Proven in Appendix A.3).

**Theorem 4.** *Given the set function $G(\mathbb{S})$ defined in Eq. (4), let $OPT$ be an optimal solution of the optimization problem (5) (or equivalently (7)). If Algorithm 1 returns $\widehat{\mathbb{S}}$ as the solution, then $G(\widehat{\mathbb{S}}) \leq G(OPT)/[(1 - \kappa_G)\alpha_G^2]$, where $\kappa_G$ and $\alpha_G$ are computed using Theorem 2*

## 4 EXPERIMENTS WITH REAL DATA

In this section, we provide a comprehensive evaluation of our method on four real world classification datasets and show that our method is able to outperform several unsupervised partitioning methods and mixture models. Next, we evaluate PRESTO in the context of resource constrained deep learning methods, where we show that PRESTO is able to provide comparable accuracy with respect to state-of-the-art models for network pruning and knowledge distillation. Appendix F contains additional experimental results.

### 4.1 EXPERIMENTAL SETUP

**Datasets.** We experiment with four real world classification datasets, *viz.*, CIFAR10 (Krizhevsky, 2009), PathMNIST (PMNIST) (Yang et al., 2021), DermaMNIST (DMNIST) (Yang et al., 2021) and SVHN (Netzer et al., 2011). Appendix D contains additional details about the datasets.

| | Accuracy $\mathbb{P}(\hat{y} = y)$ | | | | Macro F1-score | | | |
|---|---|---|---|---|---|---|---|---|
| Method | CIFAR10 | PMNIST | DMNIST | SVHN | CIFAR10 | PMNIST | DMINST | SVHN |
| Equal-Kmeans | 89.83 | 82.91 | 73.97 | 87.41 | 89.84 | 77.60 | 43.00 | 86.00 |
| Kmeans++ | 89.50 | 82.64 | 73.72 | 87.02 | 89.48 | 77.20 | 44.30 | 85.50 |
| Agglomerative | 89.35 | 81.67 | 75.46 | 87.26 | 89.34 | 75.30 | 47.00 | 85.90 |
| MoE | 88.73 | 81.29 | 72.37 | 85.68 | 88.69 | 75.57 | 37.11 | 84.23 |
| GMM | 89.66 | 83.20 | 73.72 | 87.15 | 89.65 | 77.60 | 44.30 | 85.80 |
| BGMM | 89.66 | 81.88 | 73.72 | 87.35 | 89.65 | 75.82 | 44.30 | 85.90 |
| Learn-MLR | 89.99 | 81.57 | 74.02 | 87.07 | 89.98 | 75.90 | 43.60 | 85.70 |
| **PRESTO** | 90.04 | 83.70 | 75.61 | 88.11 | 90.00 | 77.80 | 47.70 | 86.80 |

Table 1: Comparison of classification accuracy $\mathbb{P}(\hat{y} = y)$ and Macro-F1 score of PRESTO against three unsupervised partitioning methods (Equal-Kmeans (Bennett et al., 2000), Kmeans++ (Arthur & Vassilvitskii, 2006), Agglomerative (Müllner, 2011)); Mixture of Experts(MoE) (Shazeer et al., 2017) and three mixture models (GMM (Bishop & Nasrabadi, 2006), BGMM (Attias, 1999) and Learn-MLR (Pal et al., 2022)) for all datasets. In all cases, we used exactly the same model architecture for each component, which is a ResNet18 network with reduced capacity (described in Section 4.1) The best and second best results are highlighted in green and yellow respectively.

**Implementation details.** For all models, we extract features $x$ from fifth layer of a pre-trained ResNet18 (He et al., 2016). We design $h_{\theta_k}$ using a neural architecture similar to ResNet18 with reduced capacity– specifically, starting with the sixth layer and until the eighth layer, we reduce the number of convolutional filters to 4 instead of 128, 8 instead of 256 and 16 instead of 512, respectively. In each case, we set the number of model components $K$ using cross validation. We found $K = 4$ for all datasets except DMNIST and $K = 3$ for DMNIST. We use a fully connected single layer neural network to model the additional classifier $\pi_\phi$ that is used to decide the model component to be assigned to an instance during inference. The experimental setup is discussed in further details in Appendix D.

## 4.2 RESULTS

**Comparison with unsupervised partitioning methods and mixture models.** We first compare our method against several unsupervised partitioning methods and mixture models. In the case of unsupervised partitioning, we use clustering methods to slice the data into $K$ partitions before the start of the training and then use these data partitions to train the $K$ models. Specifically, we consider three unsupervised partitioning methods, *viz.*, (1) Equal-Kmeans (Bennett et al., 2000) which is a K-means clustering method that ensures equal distribution of instances across different clusters, (2) Kmeans++ (Arthur & Vassilvitskii, 2006), a variant of K-means method that uses a smarter initialisation of cluster centers. (3) Agglomerative clustering (Müllner, 2011), where the clusters are built hierarchically; (4) Mixture of Experts(MoE) (Shazeer et al., 2017), a Sparsely-Gated Mixture-of-Experts layer (MoE), which selects a sparse combination from a set of expert models using a trainable gating mechanism to process each input, and three mixture models, *viz.*, (5) Gaussian mixture models (GMM) (Bishop & Nasrabadi, 2006), (6) Bayesian Gaussian mixture model (BGMM) (Attias, 1999) and (7) Learn-MLR (Pal et al., 2022) which presents a version of the AM algorithm for learning mixture of linear regressions. We adapt this algorithm for classification as a baseline for PRESTO. Across all baselines, we employed exactly same set of model architectures. More implementation details of these methods is given in Appendix D.

In Table 1, we summarize the results in terms of classification accuracy $\mathbb{P}(\hat{y} = y)$ and Macro-F1 score. The Macro-F1 score is the harmonic mean of the precision and recall. We make the following observations. (1) PRESTO demonstrates better predictive accuracy than all the baselines. (2) For three out of four datasets, the unsupervised partitioning methods outperform the mixture models. Note that there is no consistent winner among the partitioning methods. Thus, the best choice of the underlying partitioning algorithm changes across different dataset. However, since the optimal clusters obtained by our method are guided by a supervised loss minimization, PRESTO performs well across all datasets.

**Application on resource constrained learning.** Given $K$, the number of partitions, PRESTO works with $K$ lightweight models, which can be trained on different partitions of a dataset, instead of training a very large model on the entire dataset. To evaluate the efficacy of PRESTO on a resource constrained learning setup, we compare it against two model pruning methods, *viz.*, (1) GraSP (Wang et al., 2020a), (2) SNIP (Lee et al., 2019) and two methods for knowledge distillation, *viz.*, (3) KD (Hinton et al., 2015) (4) DGKD (Son et al., 2021) in terms of both accuracy and GPU memory used during training. Pruning methods mask connections of a larger model that are relatively less

| Method | Accuracy $\mathbb{P}(\hat{y} = y)$ | | | | Training time GPU Usage (MiB) | | | |
|---|---|---|---|---|---|---|---|---|
| | CIFAR10 | PMNIST | DMNIST | SVHN | CIFAR10 | PMNIST | DMNIST | SVHN |
| GraSP | 89.37 | 85.60 | 74.71 | 89.16 | 1107.21 | 1173.46 | 1173.16 | 1173.97 |
| SNIP | 88.24 | 85.28 | 73.77 | 89.36 | 863.73 | 1149.08 | 910.49 | 910.54 |
| KD | 90.06 | 75.32 | 72.37 | 89.08 | 272.00 | 250.69 | 250.56 | 251.81 |
| DGKD | 90.17 | 76.24 | 72.02 | 88.97 | 272.00 | 250.09 | 250.56 | 252.06 |
| **PRESTO** | 90.04 | 83.70 | 75.61 | 88.11 | 173.43 | 182.24 | 58.12 | 234.48 |

Table 2: Comparison of accuracy $\mathbb{P}(\hat{y} = y)$ and maximum GPU memory used during training (in mebibytes, MiB) between PRESTO and two model pruning methods, *viz.*, GraSP (Wang et al., 2020a) and SNIP (Lee et al., 2019) and two knowledge distillation methods, *viz.*, KD (Hinton et al., 2015) and DGKD (Son et al., 2021) on 20% held-out set. Numbers in green and yellow indicate the best and the second best method.

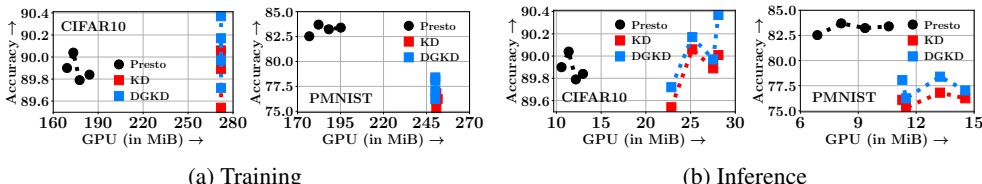

(a) Training         (b) Inference

Figure 3: Trade off between accuracy and maximum GPU memory for KD, DGKD and PRESTO during training (panel (a)) and inference (panel (b)) for CIFAR10 and PMNIST.

important for training, whereas knowledge distillation methods start with a high capacity model (teacher model) and then design a lightweight model (student) that can mimic the output of the large model with fewer parameters. For a fair comparison, we maintain roughly similar (with 5% tolerance) number of parameters of each of the final lightweight models obtained by the pruning methods and of the student models given by the knowledge distillation methods; the total number parameters used in our approach is $\sum_{k=1}^{K} \dim(\boldsymbol{\theta}_k)$

First, we set $K$ of PRESTO same as in Table 1. This gives the number of parameters of PRESTO as 82690 for CIFAR10, 54993 for PMNIST, 41047 for DMNIST and 55138 for SVHN. In Table 2, we summarize the results for this setup. We observe that PRESTO consumes significantly lesser GPU memory than any other method across all datasets, whereas it yields the best accuracy for DMNIST. For DMNIST, it consumes an impressive 77% lesser GPU memory than the second most efficient method, *viz.*, DGKD. The existing baselines use high capacity models, as reference models in pruning and teacher models in knowledge distillation, resulting in high training GPU memory consumption.

Next we vary the number of parameters $p$ of PRESTO (through $K$) as well as the baselines $p \in [62020, 124030]$ for CIFAR10, $p \in [41247, 82485]$ for PMNIST, $p \in [41047, 82087]$ for DM-NIST, $p \in [41356, 82702]$ for SVHN and probe the variation accuracy *vs.* maximum GPU memory used during both training and test. In Figure 3, we summarize the results for PRESTO and KD and DGKD— the two most resource efficient methods from Table 2. We make the following observations. (1) PRESTO consumes significantly lower memory for diverse model size during both training and inference; and, (2) In most cases, as we change the model size, the accuracies obtained by the baselines vary widely whereas, our model often shows only insignificant changes as we change the model size.

# 5 CONCLUSION

We present PRESTO, a novel framework of learning mixture models via data partitioning. Individual models specialising on a data partition, present a good alternative to learning one complex model and help achieve better generalisation. We present a joint discrete-continuous optimization algorithm to form the partitions with good approximation guarantees. With our experiments we demonstrate that PRESTO achieves best performance across different datasets when compared with several mixture-models and unsupervised partitioning methods. We also present that PRESTO achieves best accuracy *vs.* memory utilisation trade-off when compared with knowledge distillation and pruning methods. Our work opens several areas for research, such as that of handling datasets with larger class imbalance, outlier detection, handling out of distribution data, *etc.*

## 6 REPRODUCIBILITY STATEMENT

We provide a zip file containing all the source code for PRESTO as well as the various baselines we use, as a part of the supplementary material. The implementation details and machine configuration for our experiments are given in Section D of the Appendix. All our datasets are public and can be obtained easily. Users can download the datasets and use the provided code to reproduce the results presented in this paper.

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

# Appendix

## A PROOFS OF THE TECHNICAL RESULTS IN SECTION 3

### A.1 FORMAL DISCUSSION ON MATROID

**Definition 5.** *A matroid is a combinatorial structure $M := (\mathbb{V}, \mathcal{I})$ defined on a ground set $\mathbb{V}$ and a family of independent sets $\mathcal{I} \subseteq 2^{\mathbb{V}}$, which satisfies two conditions.*

*(1) If $\mathbb{S} \subseteq \mathbb{T}$ and $\mathbb{T} \in \mathcal{I}$, then $\mathbb{S} \in \mathcal{I}$.*
*(2) If $\mathbb{S} \in \mathcal{I}$ and $\mathbb{T} \in \mathcal{I}$ and $|\mathbb{T}| > |\mathbb{S}|$, then there exists a $e \in \mathbb{T} \backslash \mathbb{S}$ so that $\mathbb{S} \cup \{e\} \in \mathcal{I}$.*

From the above definition, it is clear the all the maximal independent sets have same cardinality. A maximum independent set is called the base of the matroid.

### A.2 MONOTONICITY AND $\alpha$-SUBMODULARITY OF $G$

Here we prove the claims of Theorem 2. We repeat the theorem for convenience

**Theorem (2).** *Given the set function $G(\mathbb{S})$ defined in Eq. (4) and the individual regularized loss functions $\ell$ introduced in Eq. (1), we define: $\varepsilon_{\min} = \max_{k,i} Eigen_{\min}[\nabla^2_{\boldsymbol{\theta}_k} \ell(h_{\boldsymbol{\theta}_k}(\boldsymbol{x}_i), y_i)]$, $\underline{\ell_{\min}} = \min_{i \in D} \min_{\boldsymbol{\theta}_k}[\ell(h_{\boldsymbol{\theta}_k}(\boldsymbol{x}_i), y_i) + \lambda ||\boldsymbol{\theta}_k||^2]$ and $\overline{\ell_{\min}} = \max_{i \in D} \min_{\boldsymbol{\theta}_k}[\ell(h_{\boldsymbol{\theta}_k}(\boldsymbol{x}_i), y_i) + \lambda ||\boldsymbol{\theta}_k||^2]$. Then, we have the following results:*

*(1) Monotonicity: The function $G(\mathbb{S})$ defined in Eq. (4) is monotone non-decreasing in $\mathbb{S}$.*
*(2) $\alpha$-submodularity: If $\ell(h_{\boldsymbol{\theta}_k}(\boldsymbol{x}), y)$ is $L$-Lipschitz for all $k \in \{1, .., K\}$ and that the regularizing coefficient $\lambda$ satisfies: $\lambda > -\varepsilon_{\min}$, function $G(\mathbb{S})$ is $\alpha$-submodular with*

$$\alpha \geq \alpha^* = \frac{\underline{\ell_{\min}}}{\underline{\ell_{\min}} + \frac{2L^2}{\lambda + 0.5\varepsilon_{\min}} + \frac{\lambda L^2}{(\lambda + 0.5\varepsilon_{\min})^2}} \tag{9}$$

*(3) Generalized curvature: The generalized curvature $\kappa_G(\mathbb{S})$ for any set $\mathbb{S}$ is given by: $\kappa_G(\mathbb{S}) \leq \kappa_G^* = 1 - \underline{\ell_{\min}}/\overline{\ell_{\min}}$.*

*Proof.* (1) (**Proof of monotonicity**) We define the regularized $g(\boldsymbol{\theta}_k, S) = \sum_{i \in S} \ell(h_{\boldsymbol{\theta}_k}(x_i), y_i) + \lambda ||\boldsymbol{\theta}_k||^2$. First, we show that $g(\boldsymbol{\theta}_k^*(S \cup \{j\}), S \cup \{j\}) - g(\boldsymbol{\theta}_k^*(S), S) > 0$. To that aim, note that,

$$g(\boldsymbol{\theta}_k^*(S \cup \{j\}), S \cup \{j\}) - g(\boldsymbol{\theta}_k^*(S), S)$$
$$= g(\boldsymbol{\theta}_k^*(S \cup \{j\}), S \cup \{j\}) - g(\boldsymbol{\theta}_k^*(S \cup \{j\}), S) + g(\boldsymbol{\theta}_k^*(S \cup \{j\}), S) - g(\boldsymbol{\theta}_k^*(S), S) \tag{10}$$

Now, since $\boldsymbol{\theta}_k^*(S) = \operatorname{argmin}_{\boldsymbol{\theta}_k} g(\boldsymbol{\theta}_k, S)$, we have $g(\boldsymbol{\theta}_k^*(S \cup \{j\}), S) \geq g(\boldsymbol{\theta}_k^*(S), S)$. From Eq. (10), this leads to the following inequality:

$$g(\boldsymbol{\theta}_k^*(S \cup \{j\}), S \cup \{j\}) - g(\boldsymbol{\theta}_k^*(S), S) \geq g(\boldsymbol{\theta}_k^*(S \cup \{j\}), \{j\}) \geq \underline{\ell_{\min}} > 0 \tag{11}$$

Now, if we include a new element $(j, t)$ into $\mathbb{S}$, then it will only change the loss $g(\boldsymbol{\theta}_t^*(S), S)$ among all model components. Thus we have,

$$G(\mathbb{S} \cup \{(j, t)\}) - G(\mathbb{S}) \geq g(\boldsymbol{\theta}_t^*(S \cup \{j\}), (S \cup \{j\})) - g(\boldsymbol{\theta}_t^*(S), S) > 0$$

(2) (**Proof of $\alpha$-submodularity**) First, we try to show $\alpha$-submodularity of $g(\boldsymbol{\theta}_k^*(S), S)$. Hence, we first bound the following ratio:

$$\frac{g(\boldsymbol{\theta}_k^*(S \cup \{j\}), S \cup \{j\}) - g(\boldsymbol{\theta}_k^*(S), S)}{g(\boldsymbol{\theta}_k^*(T \cup \{j\}), T \cup \{j\}) - g(\boldsymbol{\theta}_k^*(T), T)} \tag{12}$$

Eq. (11) directly provides bound on the numerator of this ratio:

$$g(\boldsymbol{\theta}_k^*(S \cup \{j\}), S \cup \{j\}) - g(\boldsymbol{\theta}_k^*(S), S) \geq \underline{\ell_{\min}} \tag{13}$$

Next, we try to upper bound $g(\boldsymbol{\theta}_k^*(T \cup \{j\}), T \cup \{j\}) - g(\boldsymbol{\theta}_k^*(T), T)$. We note that:

$$g(\boldsymbol{\theta}_k^*(T \cup \{j\}), T \cup \{j\}) - g(\boldsymbol{\theta}_k^*(T), T)$$
$$= g(\boldsymbol{\theta}_k^*(T \cup \{j\}), T \cup \{j\}) - g(\boldsymbol{\theta}_k^*(T), T \cup \{j\}) + g(\boldsymbol{\theta}_k^*(T), T \cup \{j\}) - g(\boldsymbol{\theta}_k^*(T), T). \tag{14}$$

Now, since $\boldsymbol{\theta}_k^*(T \cup \{j\}) = \operatorname{argmin}_{\boldsymbol{\theta}_k} g(\boldsymbol{\theta}_k, T \cup \{j\})$, we have:

$$g(\boldsymbol{\theta}_k^*(T \cup \{j\}), T \cup \{j\}) - g(\boldsymbol{\theta}_k^*(T), T \cup \{j\}) \leq 0 \tag{15}$$

Next, we note that $g(\boldsymbol{\theta}_k^*(T), T \cup \{j\}) - g(\boldsymbol{\theta}_k^*(T), T) = g(\boldsymbol{\theta}_k^*(T), \{j\})$ which gives us:

$$g(\boldsymbol{\theta}_k^*(T \cup \{j\}), T \cup \{j\}) - g(\boldsymbol{\theta}_k^*(T), T) = g(\boldsymbol{\theta}_k^*(T), \{j\}) \tag{16}$$

This leads us to bound the ratio in Eq. (12) as follows:

$$\frac{g(\boldsymbol{\theta}_k^*(S \cup \{j\}), S \cup \{j\}) - g(\boldsymbol{\theta}_k^*(S), S)}{g(\boldsymbol{\theta}_k^*(T \cup \{j\}), T \cup \{j\}) - g(\boldsymbol{\theta}_k^*(T), T)} \tag{17}$$

$$\geq \frac{g(\boldsymbol{\theta}_k^*(S \cup \{j\}), \{j\})}{g(\boldsymbol{\theta}_k^*(T), \{j\})}$$

$$\geq \frac{g(\boldsymbol{\theta}_k^*(S \cup \{j\}), \{j\})}{g(\boldsymbol{\theta}_k^*(S \cup \{j\}), \{j\}) + g(\boldsymbol{\theta}_k^*(T), \{j\}) - g(\boldsymbol{\theta}_k^*(S \cup \{j\}), \{j\})}. \tag{18}$$

To bound the denominator, we note that:

$$g(\boldsymbol{\theta}_k^*(T), \{j\}) - g(\boldsymbol{\theta}_k^*(S \cup \{j\}), \{j\}) \leq \ell(h_{\boldsymbol{\theta}_k^*(T)}(x_j), y_j) - \ell(h_{\boldsymbol{\theta}_k^*(S \cup \{j\})}(x_j), y_j) + \lambda \|\boldsymbol{\theta}_k^*(T)\|^2$$

Now, we note that

$$0 \geq g(\boldsymbol{\theta}_k^*(T), T) - g(\mathbf{0}, T) = \lambda \|\boldsymbol{\theta}_k^*(T)\|^2 + \left( \nabla_{\boldsymbol{\theta}} \ell(h_{\boldsymbol{\theta}}(x_i), y_i) \right)^\top_{\boldsymbol{\theta}=\mathbf{0}} \boldsymbol{\theta}_k^*(T)$$

$$+ \frac{1}{2}[\boldsymbol{\theta}_k^*(T) \nabla^2 \ell(h_{\boldsymbol{\theta}}(x_i), y_i)]_{\boldsymbol{\theta} \in (0, \boldsymbol{\theta}_k^*(T))} \boldsymbol{\theta}_k^*(T)$$

$$\implies \frac{1}{2} \|\boldsymbol{\theta}_k^*(T)\|^2 \varepsilon_{\min} + \lambda \|\boldsymbol{\theta}_k\|^2 - L \|\boldsymbol{\theta}_k\| < 0 \tag{19}$$

Thus, we have: $\|\boldsymbol{\theta}_k\| \leq \frac{L}{\lambda + \frac{\varepsilon_{\min}}{2}}$. Putting this in Eq. (19), we have:

$$g(\boldsymbol{\theta}_k^*(T), \{j\}) - g(\boldsymbol{\theta}_k^*(S \cup \{j\}), \{j\}) \leq L \|\boldsymbol{\theta}_k^*(T) - \boldsymbol{\theta}_k^*(S \cup \{j\})\| + \lambda \|\boldsymbol{\theta}_k^*(T)\|^2 \tag{20}$$

$$\leq \frac{2L^2}{\lambda + \frac{\varepsilon_{\min}}{2}} + \frac{\lambda L^2}{(\lambda + \frac{\varepsilon_{\min}}{2})^2} \tag{21}$$

Thus, replacing the above quantity in Eq. (18), we have:

$$\alpha \geq \alpha^* = \frac{\underline{\ell_{\min}}}{\underline{\ell_{\min}} + \frac{2L^2}{\lambda + 0.5\varepsilon_{\min}} + \frac{\lambda L^2}{(\lambda + 0.5\varepsilon_{\min})^2}} \tag{22}$$

(3) (**Proof of the bound on Generalized curvature**) From the definition of curvature,

$$1 - \kappa_g(S) = \min_{a \in V} \frac{g(\boldsymbol{\theta}_k^*(S), S) - g(\boldsymbol{\theta}_k^*(S \setminus \{a\}), S \setminus \{a\})}{g(\boldsymbol{\theta}_k^*(\{a\}), \{a\})} \tag{23}$$

From Eq. (11),

$$g(\boldsymbol{\theta}_k^*(S), S) - g(\boldsymbol{\theta}_k^*(S \setminus \{a\}), S \setminus \{a\}) \geq \underline{\ell_{\min}} \tag{24}$$

$$g(\boldsymbol{\theta}_k^*(\{a\}), \{a\}) \leq \max_a \left[ \lambda \|\boldsymbol{\theta}_k\|^2 + l(f_{\boldsymbol{\theta}_k}(x_a), y_a) \right] = \overline{\ell_{\min}} \tag{25}$$

Thus,

$$\kappa_g(S) \leq 1 - \frac{\underline{\ell_{\min}}}{\overline{\ell_{\min}}} \tag{26}$$

If we add an element $a = (j, t)$ to $\mathbb{S}$ only $g(\boldsymbol{\theta}_t, S)$ will be changed among the component models. Thus,

$$\kappa_G(\mathbb{S}) = 1 - \min_{a \in V} \frac{G(\mathbb{S}) - G(\mathbb{S} \setminus \{a\})}{G(a)} \tag{27}$$

$$= 1 - \min_{a \in V} \frac{g(\boldsymbol{\theta}_t^*(S), S) - g(\boldsymbol{\theta}_t^*(S \setminus \{a\}), S \setminus \{a\})}{g(\boldsymbol{\theta}_t^*(\{a\}), \{a\})}$$

$$\geq 1 - \frac{\underline{\ell_{\min}}}{\overline{\ell_{\min}}}$$

$\square$

## A.3 APPROXIMATION GUARANTEES

We next prove the approximation bound for Theorem 4.

**Theorem** (4). *If the function $G$ is $\alpha_G$-submodular and has a curvature $\kappa_G$, Algorithm 1 obtains an approximation guarantee of $\frac{|D|}{\alpha_G(1+(|D|-1)(1-\kappa_G)\alpha_G)} \leq \frac{1}{\alpha_G^2(1-\kappa_G)}$ assuming there exists a perfect training oracle in Lines (6,12,14).*

*Proof.* From the definition of $\alpha$-submodularity, note that $\alpha_G G(\mathbb{S}) \leq \sum_{i \in \mathbb{S}} G(i)$. Next, we can obtain the following inequality for any $k \in \mathbb{S}$ using weak submodularity:

$$G(\mathbb{S}) - G(k) \geq \alpha_G \sum_{j \in \mathbb{S} \setminus k} (G(j|\mathbb{S} \setminus j) \tag{28}$$

We can add this up for all $k \in \mathbb{S}$ and obtain:

$$|\mathbb{S}|G(\mathbb{S}) - \sum_{k \in \mathbb{S}} G(k) \geq \alpha_G \sum_{k \in \mathbb{S}} \sum_{j \in \mathbb{S} \setminus k} (G(j|\mathbb{S} \setminus j)$$

$$\geq \alpha_G(|\mathbb{S}| - 1) \sum_{k \in \mathbb{S}} G(k|\mathbb{S} \setminus k) \tag{29}$$

Finally, from the definition of curvature, note that $G(k|\mathbb{S} \setminus k) \leq (1 - \kappa_f)G(k)$. Combining all this together, we obtain:

$$|\mathbb{S}|G(\mathbb{S}) \geq (1 + \alpha_G(1 - \kappa_f)(|\mathbb{S}| - 1)) \sum_{j \in \mathbb{S}} G(j) \tag{30}$$

which implies:

$$\sum_{j \in \mathbb{S}} G(j) \leq \frac{|\mathbb{S}|}{1 + \alpha_G(1 - \kappa_G)(|S| - 1)} G(\mathbb{S}) \tag{31}$$

Combining this with the fact that $\alpha_G G(\mathbb{S}) \leq \sum_{i \in \mathbb{S}} G(i)$, we obtain that:

$$G(\mathbb{S}) \leq \frac{1}{\alpha_G} \sum_{i \in \mathbb{S}} G(i) \leq \frac{|\mathbb{S}|}{\alpha_G(1 + \alpha_G(1 - \kappa_G)(|S| - 1))} G(\mathbb{S}) \tag{32}$$

Note that $|\mathbb{S}|/(1 + \alpha_G(1 - \kappa_G)(|\mathbb{S}| - 1) \leq 1/\alpha_G^2(1 - \kappa_G)$ so we just use this factor in the approximation bound. The approximation guarantee then follows from some simple observations. In particular, given an approximation

$$m^G(\mathbb{S}) = \frac{1}{\alpha_G} \sum_{i \in \mathbb{S}} G(i) \tag{33}$$

which satisfies $G(\mathbb{S}) \leq m^G(\mathbb{S}) \leq \beta_G G(\mathbb{S})$, we claim that optimizing $m^G$ essentially gives a $\beta_G$ approximation factor. To prove this, let $\mathbb{S}^*$ be the optimal subset, and $\hat{\mathbb{S}}$ be the subset obtained after optimizing $m^G$. The following chain of inequalities holds:

$$G(\hat{\mathbb{S}}) \leq m^G(\hat{\mathbb{S}}) \leq m^G(\mathbb{S}^*) \leq \beta_G G(\mathbb{S}^*) \tag{34}$$

This shows that $\hat{\mathbb{S}}$ is a $\beta_G$ approximation of $\mathbb{S}^*$. Finally, note that this is just the first iteration of PRESTO, and with subsequent iterations, PRESTO is guaranteed to reduce the objective value since we only proceed if there is a reduction in objective value. □

## B    Comparison with EM Algorithm

General EM algorithm usually considers a prior and computes the membership probability of each datapoint at each iteration. It makes assumption about the generative mechanism of the data as well as the prior about the cluster probabilities. Such a probabilistic approach naturally leads one to develop EM algorithm.

On the other hand, our method makes no assumption about the data as well as cluster membership. We do not make any probabilistic assumption which naturally led us to develop a set optimization problem— we do not make use of any "continuous" or "soft" scores on cluster membership. Thus, our method is an MM algorithm which is an iterative set optimization algorithm, which is functionally very different from EM algorithm.

## C    Difference with Mixture of Linear Regression

All our experiments consider classification tasks. Therefore, we modify Learn-MLR  (Pal et al., 2022) to a mixture of classifiers. However, we note that the optimization *problem formulation* of the mixture of linear regression (or classification) can be viewed as special case of the *problem formulation* of our method.

In most cases, the existing methods make assumptions about the generative mechanism of the cluster membership (e.g., some definite prior) and resort to an algorithm which makes an involved use of that assumption and is significantly tailored to a particular task (mixture of regression or classification).

In contrast, our framework operates on a generic framework and does not make any specific assumption about the data or the cluster membership. Moreover, the algorithm is very different from what is used by the usual mixture model learning algorithms.

## D    Additional experimental setup

### D.1    Dataset details

We experiment with four real world classification datasets in our experiments,

- **CIFAR10**  (Krizhevsky, 2009) has images of 10 different real-world objects.
- **SVHN**  (Netzer et al., 2011) has real-world images for the 10 digits classification task
- **PathMNIST** (PMNIST)  (Yang et al., 2021) has images of colorectal cancer histology slides.
- **DermaMNIST** (DMNIST)  (Yang et al., 2021) has dermatoscopic images of common pigmented skin lesions.

Table 4 lists the further details of these datasets. We see that CIFAR10 is a balanced dataset where as PathMNIST, DermaMNIST and SVHN have large skew in their label proportions.

| Dataset | #Classes | #Instances | #Train | #Test | #Avg points per class |
|---------|----------|------------|--------|-------|----------------------|
| CIFAR10 | 10 | 60,000 | 50,000 | 10,000 | $5,000 \pm 0$ |
| SVHN | 10 | 99,289 | 73,257 | 26,032 | $7325.7 \pm 2800.66$ |
| PMNIST | 9 | 107,180 | 89,996 | 7,180 | $9,999.56 \pm 1592.4$ |
| DMNIST | 7 | 10,015 | 7,007 | 2,005 | $1,001 \pm 1,530.9$ |

Table 4: Dataset statistics

Using a ResNet18  (He et al., 2016) model whose weights are initialised with weights obtained while training with Imagenet (Deng et al., 2009) dataset, we get feature embedding of these datasets. For CIFAR10 dataset we extract embeddings from the fourth last layer of pretrained ResNet18 and for other datasets we extract embeddings from the fifth last layer.

### D.2 MODEL ARCHITECTURE USED FOR OUR METHOD, THE UNSUPERVISED PARTITIONING METHODS AND THE MIXTURE MODELS

In case of our method, the unsupervised partitioning methods and the mixture models (all methods in Table 1 and PRESTO in Table 2), we use a relatively light model. This model is exactly same in terms of architecture design to the final five layers of ResNet18 for PMNIST, DMNIST and SVHN, final four layers of ResNet18 for CIFAR10. This model design is chosen because we use extracted embeddings for training. The model reduces the convolution filters at various junctures. The fifth last layer has a filter of size 128 which is replaced with a filter of size 4. Similarly, the filter at fourth last layer is changed from 256 to 8, filter at third last layer is changed from 512 to 16. We train an additional classifier. This classifier is a single layer fully connected feed forward network that aids in predicting the final class for each instance.

### D.3 IMPLEMENTATION DETAILS FOR PRESTO

**Number of partitions $K$.** Number of partitions ($K$) is found for various datasets by performing an analysis of classification accuracy v/s $K$ in the validation. Table 5 shows $K$ for each dataset.

**Initialization.** PRESTO is initialised using one of the clustering techniques depending on the dataset. We also try to ensure that the clusters are of approximately the same size. We impose $\frac{|N|}{|K|}$ size constraints for each of the bin with some leeway. The leeway granted for every dataset is also a tunable hyperparameter.

**Final selection of $(i, k)$ from the matrix $M$.** The values of the matrix $M$ computed in line 19 of Algorithm 1 are supplemented with euclidean distance ($D$) of a point to the existing bin center. The final criterion for binning becomes $M + \epsilon D$. Here, $\epsilon = 3e - 3$ for CIFAR10, SVHN and PMNIST. However, for DMNIST, we have $\epsilon = 6e - 3$.

**Optimizer.** We use an Adam optimizer with the Cross-Entropy Loss to train the PRESTO models and the classifier. The learning rate is set to $0.01$. Table 5 encapsulates the various hyperparameters discussed for the datasets. We set the value of the regularizer $\lambda = 1e - 4$.

| Dataset | $K$ | Initialization | Leeway |
|---------|-----|----------------|--------|
| CIFAR10 | 4 | Equal-KMeans | 0 |
| SVHN | 4 | Equal-KMeans | 3000 |
| PMNIST | 4 | Equal-KMeans | 3000 |
| DMNIST | 3 | KMeans++ | 500 |

Table 5: Dataset specific hyperparameters for PRESTO

In Algorithm 1, Train($\cdot$) train the models on the respective data partitions for 10 epochs. We set `Iterations` $= 30$. The additional classifier $\pi_\phi$ is also trained for 10 epochs.

Table 1 mentions various baselines. Details about each baseline is covered in the following points.

### D.4 IMPLEMENTATION DETAILS ABOUT THE UNSUPERVISED PARTITIONING METHODS AND MIXTURE MODELS

**Equal-Kmeans.** We use the (Josh Levy-Kramer, 2022) implementation of Constrained K means clustering. We run the algorithm for $K$ clusters, and constrain the size of a cluster to be within $\left[\frac{|N|}{|K|} - \text{Leeway}, \frac{|N|}{|K|} + \text{Leeway}\right]$, i.e. roughly equally sized clusters with tolerance equal to the Leeway parameter. A Kmeans++ initialization is used for selecting the initial centroids. We keep the number of times the algorithm will be run with different centroid seeds as n_init = 10. The final result is the best output of n_init consecutive runs in terms of inertia. The algorithm runs either a max of 300 iterations or convergence, whichever occurs earlier. The convergence criterion is defined as the Frobenius norm of the difference in the cluster centers of two consecutive iterations becoming less than $1e - 4$.

**Kmeans++.** We use the scikit-learn (Pedregosa et al., 2011) implementation of Kmeans++ with the following parameters:
The number of clusters is set as $K$. A Kmeans++ initialization is used for selecting the initial

centroids. We keep the number of times the algorithm will be run with different centroid seeds as n_init = 10. The final result is the best output of n_init consecutive runs in terms of inertia. The algorithm runs either a max of 300 iterations or convergence, whichever occurs earlier. The convergence criterion is defined as the Frobenius norm of the difference in the cluster centers of two consecutive iterations becoming less than $1e - 4$.

**Agglomerative.** It is a bottom-up hierarchical clustering approach. We use the scikit-learn (Pedregosa et al., 2011) implementation of Agglomerative We use the *Euclidean* distance metric to calculate distance between data points. Initially each data point is in its own cluster. The clusters are subsequently merged based on a linkage criterion. We use the ward linkage criterion, which minimizes the sum of square differences within each cluster. The number of clusters is set as $K$.

**GMM.** We use the scikit-learn (Pedregosa et al., 2011) implementation of GMM with the number of components $= K$. The GMM weights are initialized according to a KMeans initialization. The convergence threshold for the EM iterations is set as $1e - 3$.

**BGMM.** We use the scikit-learn (Pedregosa et al., 2011) implementation of BMM with the number of components $= K$. The BGM weights are initialized according to a KMeans initialization. The convergence threshold for the EM iterations is set as $1e - 3$. The weight concentration prior is set as dirichlet process.

**Learn-MLR.** We adapt the original paper (Pal et al., 2022) for the purpose of classification. The initial partitioning is random and the we do not have a leeway when the data is re-partitioned. The $\epsilon$ parameter which gives a contribution to the euclidean distance ($D$) is set to 0. An Adam optimizer is used with Cross-Entropy loss. The learning rate is set to $0.01$ and the regularizer $\lambda = 1e - 4$.

**Mixture of Experts.** We use the Sparse-MoE layer described in (Shazeer et al., 2017) for classification. The expert networks have the architecture described in Section D.2. We adapt the (Rau, 2019) implementation for this purpose. We use Noisy Top-K Gating as described in (Shazeer et al., 2017), which balances the number of training examples each expert receives. We set the number of experts used for each batch element, $k = 4$.

### D.5 Implementation details for resource-constrained experiments

We evaluate PRESTO in a resource-constrained learning setting by comparing with model pruning and knowledge distillation methods. The specifications for them are mentioned in the following points.

**SNIP (Lee et al., 2019).** SNIP is a pruning at initialization method, that given a large reference network prunes network connections to a desired sparsity level (i.e. number of non-zero weights). Pruning is done *prior to training*, in a data-dependent way based on the loss function at a variance scaling initialization, to create a sparse subnetwork which is then used for inference. We adapt the implementation from (Su et al., 2020) to run experiments using SNIP. We train a large reference model, with the same architecture as the PRESTO models described in Section D.2. The reference model has the convolution filter size at the fifth last layer as 256, at the fourth last layer as 512 and at the third last layer as 1024. The pruning ratio is set so that the number of nonzero parameters in the reference network after pruning are equal to that of the corresponding PRESTO model in all experiments. The models are trained on using an Adam optimizer and the Cross-Entropy loss with a learning rate of $0.1$. The reference models are trained for 300 epochs, and the subsequent pruned models are trained for a total of 20 epochs till convergence.

**GraSP (Wang et al., 2020a).** GraSP is a pruning at initialization method similar to SNIP and also learns a smaller subnetwork given a reference network, which can subsequently be trained independently. The gradient norm *after pruning* is the pruning criterion in GraSP, and those weights whose removal will result in least decrease in the gradient norm after pruning are pruned. We adapt the implementation from (Su et al., 2020) to run experiments using SNIP. We train a large reference model, with the same architecture as the PRESTO models described in Section D.2. The reference model has the convolution filter size at the fifth last layer as 256, at the fourth last layer as 512 and at the third last layer as 1024. The pruning ratio in all experiments is set so that the number of nonzero parameters in the reference network after pruning are equal to that of the corresponding PRESTO model in all experiments. The models are trained on using an Adam optimizer and the

Cross-Entropy loss with a learning rate of $0.1$. The reference models are trained for 300 epochs, and the subsequent pruned models are trained for a total of 20 epochs till convergence.

**KD (Hinton et al., 2015).** We train two models, a teacher model and a student model. The architecture of these models is same as the PRESTO models described in Section D.2. The teacher model is heavier in size, the convolution filter size at the fifth last layer is 128, at the fourth last layer is 256 and at the third last layer is 512. On the other hand, the lighter student model has convolution filter size set to 12,16 and 32. The models are trained on using an Adam optimizer and the Cross-Entropy loss. The KD Temperature hyperparameter is set to 5. The teacher model is trained and the loss of the student model is modified to include a contribution from the teacher model. The multiplicative hyperparameter controlling the contribution of the teacher model is set to 0.9.

**DGKD (Son et al., 2021).** This method requires us to train 3 models, a teacher model, a TA model and a student model. The teacher model is the heaviest, having the convolution filters set to 128, 256 and 512. The TA model is slightly lighter with filter size of 32, 64 and 128. The student model is the lightest with filter size of 12, 16 and 32. The temperature hyperparameter is 5 and the factor controlling the contribution of the teacher and TA model is set at 0.9. We train the teacher and the TA models independently. As in KD, the loss function of the student model is tweaked to consider a contribution from the teacher and the TA models.

**Machine configuration.** We performed our experiments on a computer system with Ubuntu 16.04.6 LTS, an i-7 with 8 cores CPU and a total RAM of 528 GBs. The system had a single Titan RTX GPU which was employed in our experiments.

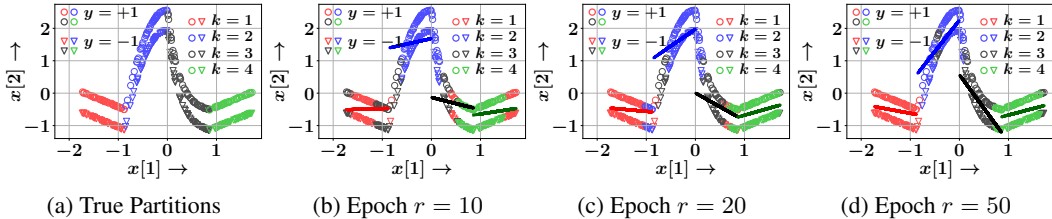

| (a) True Partitions | (b) Epoch $r = 10$ | (c) Epoch $r = 20$ | (d) Epoch $r = 50$ |

Figure 6: Snapshot of the true partition (panel (a)); and, the snapshots of the trained models $\{h_{\boldsymbol{\theta}_k} \mid k \in [K]\}$ and the partitions $\{S_k \mid k \in [K]\}$ predicted by PRESTO (panels (b)–(d)) on the synthetic dataset during progression of Algorithm 1. The dataset is generated using a degenerate mixture model with $K = 4$ components. Here, an instance $(\boldsymbol{x}_i, y_i)$ belongs to exactly one of the four sets $\{S_k^* \mid k \in [4]\}$ with probability 1, where $S_k^*$ are defined in Eqs. (35)— (38). Each model component is an SVM, *i.e.*, $h_{\boldsymbol{\theta}_k}(\boldsymbol{x}) = \boldsymbol{w}_k^\top \boldsymbol{x} + b_k$ and $\ell(\hat{y}, y) = (1 - \hat{y}y)_+$. We observe that as $r$ increases, PRESTO becomes more and more accurate in assigning an instance to the correct mixture component and finally, at $r = 50$, it is almost able to recover the true mixture component— the ground truth assignments of the instances (panel (a)) is extremely close to the final assignments (panel (d)).

# E EXPERIMENTS ON SYNTHETIC DATA

In this section, we experiment with synthetically generated instances from a latent degenerate mixture distribution and show that, Algorithm 1 can accurately recover the partitions which correspond to the true mixture components.

## E.1 EXPERIMENTAL SETUP

**Dataset generation.** We generate $|D| = 20000$ examples $\{(\boldsymbol{x}_i, y_i)\}_{i \in D}$ where $\mathcal{X} = [-1.73, 1.73] \times [-1.17, 2.59]$ and $\mathcal{Y} = \{-1, +1\}$. The instances $(\boldsymbol{x}, y)$ belong to one of the $K = 4$ sets, *viz.*, $S_1^*, S_2^*, S_3^*, S_4^*$, defined as follows:

$$S_1^* = \{(\boldsymbol{x}, y) \mid 7x[1] + x[2] - 2y - 27 = 0 \wedge \boldsymbol{x} \in [-1.73, -0.86] \times [-1.16, 0.04]\} \tag{35}$$

$$S_2^* = \{(\boldsymbol{x}, y) \mid 20(x[1]^2) - 200x[1] + 414 + x[2] - 2y \wedge \boldsymbol{x} \in [-0.87, 0.01] \times [-1.15, 2.59]\} \tag{36}$$

$$S_3^* = \{(\boldsymbol{x}, y) \mid x[2] + 1 + \exp(-x[1]) + 2y \wedge \boldsymbol{x} \in [-0.01, 0.87] \times [-1.17, 2.56]\} \tag{37}$$

$$S_4^* = \{(\boldsymbol{x}, y) \mid 7x[1] + x[2] - 2y + 43 = 0 \wedge \boldsymbol{x} \in [0.86, 1.73] \times [-1.17, 0.04]\} \tag{38}$$

Panel (a) of Figure 6 shows a scatter plot of $(\boldsymbol{x}, y)$. The instances are homogeneously distributed across all components, *i.e.*, $|S_k^*| = 5000$.

**Choice of $f_{\boldsymbol{\theta}_k}$ and $\ell$.** We consider $K = 4$ linear support vector machines $h_{\boldsymbol{\theta}_k}(x) = \boldsymbol{w}_k^\top \boldsymbol{x} + b_k$ with $\boldsymbol{\theta}_k = \{\boldsymbol{w}_k, b_k\}$ set $\ell$ as the margin based hinge loss, *i.e.*, $\ell(h_{\boldsymbol{\theta}_k}(\boldsymbol{x}), y) = (1 - y(\boldsymbol{w}_k \boldsymbol{x}_k + b_k))_+$. Then we apply PRESTO (Algorithm 1) to simultaneously learn the parameters $\{(\boldsymbol{w}_k, b_k) \mid 1 \le k \le 4\}$ and the partitions $S_k$ such that $D = \cup_{k=1}^4 S_k$. In addition to the classification accuracy, we also measure the aggregated error in predicting the correct mixture component which is defined as $\text{Err}_S = \sum_{i,j} \mathbf{1}(i, j \in S_k^*, i \in S_t, j \in S_{t'} \text{ with } t \neq t')$. Thus, $\emptyset$ computes the number of pairs that belong to the same latent mixture component $S_k^*$, but are predicted to have different mixture components by PRESTO.

## E.2 RESULTS

In Figure 6, we plot different snapshots of the models $h_{\boldsymbol{\theta}_k}$ and the partitions $D = \cup_{k=1}^4 S_k$, as PRESTO progresses during different iterations $r$ (line 9 in Algorithm 1). We make the following observations: (1) As $r$ increases, the classification accuracy increases as well as mixture component prediction error $\text{Err}_S$ decreases. For $r = 50$, we observe that $\text{Err}_S$ becomes very small and PRESTO is able to recover the true partitions, *i.e.*, $S_k^* \approx S_k$ (2) We observe that PRESTO finds it relatively difficult to correctly assign the instances on $S_2^*$ and $S_3^*$ to the right mixture components. This is due to two reasons: first, the instances on $S_2^*$ and $S_3^*$ fit naturally to a nonlinear SVMs, whereas we attempt to fit a linear SVM; second, the instances in $S_2^*$ and $S_3^*$, which are around $(0, 0)$, are close to each other— this poses difficulty in demarcating their model components.

## F   ADDITIONAL EXPERIMENTS ON REAL DATA

**Accuracy vs $K$.** Here, we aim to assess the impact of number of partitions ($K$) on the accuracy. $K$ is varied from 3 to 6. Figure 7 shows the variation of $K$ with the test accuracy and the Macro-F1 score. We observe that the performance is stable across these range.

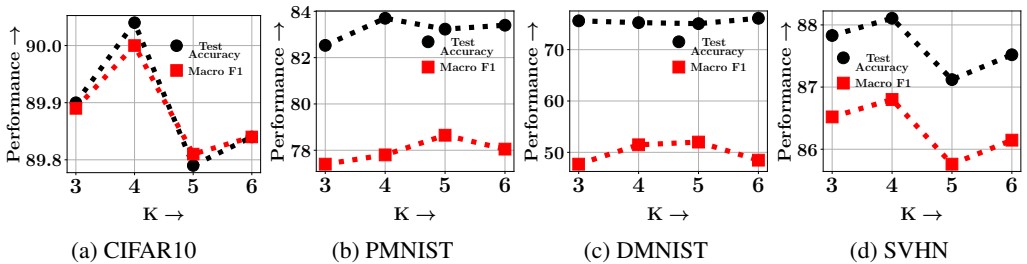

Figure 7: Accuracy vs. the number of partitions $K$

**Accuracy vs GPU usage for DMNIST and SVHN.** Here, we perform the same experiments on DMNIST and SVHN as Figure 3. Figure 8 summarizes the results, which reveal same insights as Figure 3.

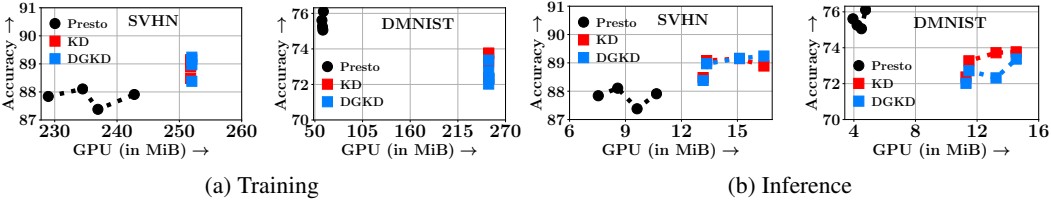

Figure 8: Trade off between accuracy and maximum GPU memory for KD, DGKD and PRESTO during training (panel (a)) and inference (panel (b)) for DMNIST and SVHN.

## G   EXPLANATION OF PRESTO (ALGORITHM 1)

In PRESTO we aim to *minimize* a monotone approximate-submodular function $G$. In contrast to submodular or approximate submodular maximization where one can use greedy algorithms, *minimizing* the approximate submodular function $G$ needs a completely different approach. Here, we resort to minimizing an upper bound $m$ of $G$— reducing the value of this upper bound $m$ will ensure the value of the underlying function $G$ remains low. Now, such an upper bound $m$ has to be such that the gap between $m$ and $G$ is low and maximizing $m$ is convenient. To this aim, we choose $m$ to be modular. One can connect such a modular approximation of a set function to a simple linear approximation of a complex nonlinear function in the context of continuous optimization.

Now minimizing the modular function $m_{\hat{\mathbb{S}}}[\mathbb{S}]$ wrt $\mathbb{S}$ is equivalent to taking the minimum of $[m_{\hat{\mathbb{S}}}[(i,k)]$ over different $(i,k)$ pairs ( since, $modular(Set) = \sum_{e \in Set} modular(e)$). In each iteration of Algorithm 1, we simply compute the minimum of $m$ and update $\mathbb{S}$ accordingly. Note that:

$$m_{\hat{\mathbb{S}}}^{G}[\mathbb{S}] = G(\hat{\mathbb{S}}) - \sum_{(i,k) \in \hat{\mathbb{S}}} \alpha_G G((i,k)|\hat{\mathbb{S}} \setminus \{(i,k)\}) + \sum_{(i,k) \in \hat{\mathbb{S}} \cap \mathbb{S}} \alpha_G G((i,k)|\hat{\mathbb{S}} \setminus \{(i,k)\}) \quad (39)$$

$$+ \sum_{(i,k) \in \mathbb{S} \setminus \hat{\mathbb{S}}} \frac{G((i,k)|\emptyset)}{\alpha_G}$$

Now, maximizing $m_{\hat{\mathbb{S}}}^{G}[(i,k)]$ is same as maximizing $\alpha_G G((i,k)|\hat{\mathbb{S}} \setminus \{(i,k)\})$ or $\frac{G((i,k)|\emptyset)}{\alpha_G}$ depending on whether $(i,k) \in \hat{\mathbb{S}}$ (3nd term in Eq (39)) or $(i,k) \notin \hat{\mathbb{S}}$ (4th term in Eq (39)). We compute these two quantities in L15 (line no. 15 in Algorithm 1) and L17 of the algorithm and store the values of

m at different pairs $(i, k)$ in the matrix M. Finally, we take the minimum of M to find $(i^*, k^*)$ (L19) which indicates that the partition $k^*$ should contain $i^*$. Therefore, we include $i^*$ to $\hat{S}_{k^*}$ (L20). Now, since any instance can belong to exactly one partition, we remove $i^*$ from all other partition $k \neq k^*$ (L21). Finally, we update $\hat{\mathbb{S}}$ (L22).

## H  ADDITIONAL EXPERIMENTS WITH DATASETS HAVING LABEL NOISE

The datasets are homogenous, extremely balanced and there is a lack of noise. In this section, we present experiments where we added different amounts of label noise. Specifically, we changed each label $y$ to a wrong label $y^{'}$ with probability 10% and then probe the performance. The results for CIFAR10 and DMNIST are presented in Table 9. We observe that PRESTO performs much better than the clustering baselines.

| Method | Accuracy $\mathbb{P}(\hat{y} = y)$ | |
| | CIFAR10 | DMNIST |
| --- | --- | --- |
| PRESTO | 89.01 | 75.31 |
| EqKMeans | 88.49 | 74.46 |
| Kmeans++ | 88.08 | 74.11 |
| Agglomerative | 87.59 | 74.46 |

Table 9: Comparison of classification accuracy $\mathbb{P}(\hat{y} = y)$ of PRESTO against three unsupervised partitioning methods (Equal-Kmeans (Bennett et al., 2000), Kmeans++ (Arthur & Vassilvitskii, 2006), Agglomerative (Müllner, 2011)). We change each label $y$ to a wrong label $y^{'}$ with probability 10%, thereby adding heterogenity to the dataset. Numbers in green and yellow indicate the best and the second best method.

## I  TIME COMPLEXITY ANALYSIS

Although PRESTO performs data partitioning while simultaneously learning a set of mixture of models, the partitioning time (lines 19-22 in Algorithm 1) is negligible as the bulk of the time is devoted to training (lines 10-18 in Algorithm 1). Since the training stage is common to all the methods, PRESTO does not provide much disadvantage in terms of the time. Table 10 presents an analysis of the per-iteration training time.

| Method | Training time per iteration | | | |
| | CIFAR10 | PMNIST | DMNIST | SVHN |
| --- | --- | --- | --- | --- |
| PRESTO | 9.6 min | 16.6min | 1.65 min | 14.6 min |
| EqKMeans | 9.4 min | 16.6min | 1.49 min | 14.52 min |
| Kmeans++ | 7.96 min | 16.6min | 1.43 min | 14.46 min |

Table 10: Comparison of training time per iteration of PRESTO against Equal-Kmeans (Bennett et al., 2000) and Kmeans++ (Arthur & Vassilvitskii, 2006)

