# OpenReview forum: "Learning Mixture Models with Simultaneous Data Partitioning and Parameter Estimation"
_ICLR.cc/2023/Conference — Submitted to ICLR 2023_

### Official Review · Reviewer_jcju · 2022-10-21

**Confidence:** 4
**Correctness:** 2
**Technical Novelty And Significance:** 3
**Empirical Novelty And Significance:** 2
**Recommendation:** 5

**Clarity, Quality, Novelty And Reproducibility:**

The paper is well written in general. The idea is novel though there are some claims need to be better explained or justified. The experiments seem to be reproducible based on the description in the paper.




**Strength And Weaknesses:**

**Strengths**
The paper is well written and can be easily followed.

**Weakness**
1. One major concern of the reviewer is that the difference of the proposed method and mixture of linear regression is not highlighted.
- In Section 2.3, the authors mentioned that mixture of linear regression is a special case of theirs. If so, then why you get different results for these two methods in Table 1.
- Can the authors please clatify what's the difference between the proposed approach and the general EM algorithm used in fitting mixture of regression models.

2. Numerical algorithm. An MM algorithm is proposed to minimize $G(\mathbb{S})$. The success of MM algorithm depends heavily on the majorization function, which the reviewer believe shoudl be $m_{\hat{\mathbb{S}}}^{G}[\mathbb{S}]$. However, the reviewer don't think $m_{\hat{\mathbb{S}}}^{G}[\mathbb{S}]$ majorizes $G(\mathbb{S})$ since the equality does not hold when $\mathbb{S}=\hat{\mathbb{S}}$. Can the author please clarify?

3. Experiment result.
- Are the differences of different approaches in Table 1 significant?
- The k-means approaches does not use any information about label $y$ but still performs as good as the proposed method, can the authors give some justificaitons on why this is the case?


**Minor issues**

1. The definitions of monotonity and $\alpha$-submodular need to be double checked. The current version is not correct since the $f$ does not appear anywhere in the definition.
2. The definition of relizable and non-relizable is not clear.
3. The number of paritions is the same as the number of classes in the dataset, what happens if the number of partitions is overspecified or under specified?


**Summary Of The Paper:**

When the observed data comes from a population that consists of several subpopulations, it is useful to model each subpopulation separately. Hence, the paper proposes a method that simultaneously learns a partition of the dataset so that each subset of the data comes from the same distribution. The model parameters and the partitions are learned simultaneouly, which is formulated as an optimization problem. One major contribution of the paper is to design a numerical algorithm.

**Summary Of The Review:**

One major concern of the reviewer is that the difference of the proposed method and mixture of linear regression.  The paper is marginally below the acceptance threshold before the reviewer's comments are clarified.

---

> ### Author Response · Authors · 2022-11-18
> **Response to Reviewer jcju (Part-1/2)**
>
> We would like to thank the reviewer for the comments, which would indeed improve our paper.
>
> >*One major concern of the reviewer is that the difference of the proposed method and mixture of linear regression is not highlighted.
> In Section 2.3, the authors mentioned that mixture of linear regression is a special case of theirs. If so, then why you get different results for these two methods in Table 1.*
>
> We would like to first clarify in principle, all our experiments consider classification tasks. Therefore, we had to modify a mixture of linear regressors to a mixture of classifiers. However, we would like to note that the optimization *problem formulation* of the mixture of linear regression (or classification) can be viewed as special case of the *problem formulation* of our method. The PRESTO algorithm differs from the algorithm used for mixture of linear regression.
>
> However, in most cases, the existing methods make assumptions about the generative mechanism of the cluster membership (e.g., some definite prior) and resort to an algorithm which makes an involved use of that assumption and is significantly tailored to a particular task (mixture of regression or classification).
>
> In contrast, our framework operates on a generic framework and does not make any specific assumption about the data or the cluster membership. Moreover, the algorithm is very different from what is used by the usual mixture model learning algorithms.
>
> We updated this discussion in Appendix C.
>
> >*Can the authors please clarify what's the difference between the proposed approach and the general EM algorithm used in fitting mixture of regression models.*
>
> General EM algorithm usually considers a prior and computes the membership probability of each datapoint at each iteration. It makes assumption about the generative mechanism of the data as well as the prior about the cluster probabilities. Such a probabilistic approach naturally leads one to develop EM algorithm.
>
> On the other hand, our method makes no assumption about the data as well as cluster membership. We do not make any probabilistic assumption which naturally led us to develop a set optimization problem— we do not make use of any “continuous” or “soft” scores on cluster membership. Thus, our method is an MM algorithm which is an iterative set optimization algorithm, which is functionally very different from EM algorithm.
>
> We updated this discussion in Appendix B.
>
> >*Numerical algorithm. An MM algorithm is proposed to minimize $G(\mathbb{S})$. The success of MM algorithm depends heavily on the majorization function, which the reviewer believe should be $m^G_{\hat{\mathbb{S}}}[\mathbb{S}]$. However, the reviewer don't think $m^G_{\hat{\mathbb{S}}}[\mathbb{S}]$ majorizes $G(\mathbb{S})$ since the equality does not hold when $\mathbb{S} = \mathbb{\hat{S}}$. Can the author please clarify?*
>
> There may be some misunderstanding.  Note that:
> $m^G   _{\hat{\mathbb{S}}}[\mathbb{S}]=     G (\hat{\mathbb{S}}) -  \sum _{(i,k) \in \hat{\mathbb{S}}} \alpha _G G((i,k)| \hat{\mathbb{S}} \backslash \set{(i,k)}) + \sum _{(i,k)  \in \hat{\mathbb{S}} \cap \mathbb{S}}  \alpha _G G((i,k)| \hat{\mathbb{S}} \backslash \set{(i,k)}) +  \sum _{(i,k) \in \mathbb{S} \backslash \hat{\mathbb{S}}}  \frac{G((i,k) |\emptyset)}{\alpha _G} $
>
> If $\hat{\mathbb{S}} = \mathbb{S}$ , the second and third term cancels and the fourth term becomes zero. Thus, $m^G   _{\hat{\mathbb{S}}}[\mathbb{S}]=     G (\hat{\mathbb{S}})$.

---

> > ### Author Response · Authors · 2022-11-18
> > **Response to Reviewer jcju (Part-2/2)**
> >
> > [continued from Part-1]
> >
> > >*Are the differences of different approaches in Table 1 significant? The k-means approaches does not use any information about label y but still performs as good as the proposed method, can the authors give some justificaitons on why this is the case?*
> >
> > Indeed, the performance between the second best baseline and our method is close. However, note that there is no consistent winner among the baselines across all datasets. Hence, the choice of the optimal clustering baselines differs across different datasets. In this context, our method provides optimal clustering. From a practitioner’s  viewpoint, one does not have to check which clustering algorithm to apply for partitioning. Our proposal provides a meta algorithm which can find these clusters— it can automatically find which clustering protocol one should adopt for the underlying dataset.
> >
> > The reason behind the marginal gain of our algorithm is lack of heterogeneity in the datasets. The datasets are extremely balanced and there was a lack of noise. Thus, during rebuttal times, we performed experiments where we added different amounts of label noise. Specifically, we changed each label y to a wrong label y’ with probability 10%  and then probe the performance. Results are as follows for the CIFAR10 and DermaMNIST Dataset.
> >
> > | Method | CIFAR10 | DermaMNIST |
> > | - | - | - |
> > | PRESTO       | 89.01 | 75.31
> > | EqKMeans     | 88.49 | 74.46
> > | KMeans++      |  88.08 | 74.11
> > | Agglomerative |  87.59 | 74.46
> >
> > We updated these results in Appendix H.
> >
> > >*The definitions of monotonity and α-submodular need to be double checked. The current version is not correct since the f does not appear anywhere in the definition*
> >
> > We have addressed this and replaced f with G in the revised version.
> >
> > >*The number of partitions is the same as the number of classes in the dataset, what happens if the number of partitions is overspecified or under specified?*
> >
> > The number of partitions is not the same as the number of classes in the dataset. The number of partitions is a hyperparameter. In appendix F, in figure 7 we report the effect on test accuracy as we vary the number of partitions.

---

> > > ### Comment · Reviewer_jcju · 2022-12-09
> > > **post-rebuttal**
> > >
> > > Thanks for the response and the clarification on the majorization function. The idea and the algorithm seem novel, but the reviewer's major concern on the difference between the proposed method and the existing approaches remains after the rebuttal. The authors argued that the advantage of the proposed method is that it does not have any assumptions on the generative mechanism of the data. However, this comes with a cost of an additional classifier during the inference time. This plus the empirical results (both the ones in the paper and the new ones during the rebuttal) do not fully convince me to use the proposed method if I had such a problem at hand. Therefore, it would be great if the authors can dig deeper to convince the readers the advantage of the proposed method.

---

### Official Review · Reviewer_f3vu · 2022-10-22

**Confidence:** 4
**Correctness:** 3
**Technical Novelty And Significance:** 3
**Empirical Novelty And Significance:** 3
**Recommendation:** 5

**Clarity, Quality, Novelty And Reproducibility:**

The paper proposes an interesting formulation and includes approximation results.
However, it lacks in terms of clarity, especially in what concerns the 'outline of the PRESTO algorithm' (section 3.2) and the corresponding pseudocode (Algorithm 1).
In what concerns reproducibility, the authors provide the code of PRESTO along with details on the used datasets.

**Strength And Weaknesses:**

Stength
 - An interesting problem formulation is presented.
 - Partitioning is obtained through explicit specification of the partitions without resorting to a clustering model.
 - Theoretical bounds are provided.
 - An approximate solution to the constrained optimization problem is proposed.

Weaknesses
  - Method presentation (section 3.2) and algorithm description are not clear, thus they are difficult to comprehend.
  - The method can not directly provide the partition assignments of new examples. Therefore an additional classification
    model should be trained for the assignment task.
  - Evaluation is insufficient: PRESTO is not compared to various methods belonging to the Mixture of Experts category.
  - Time complexity is not reported

**Summary Of The Paper:**

The paper proposes a framework (called PRESTO) for simultaneously partitioning the data space and learning a separate model in each partition. The framework relies on solving an optimization problem with both discrete variables (data partitioning) and continuous parameters
(model parameters) using a constrained optimization method.

**Summary Of The Review:**

The paper proposes a framework (called PRESTO) for simultaneously partitioning the data space and learning a separate model in each partition. The framework relies on solving an optimization problem with both discrete variables (data partitioning) and continuous parameters (model parameters) using a constrained optimization method.

The idea is interesting and supported by theoretical results.

However, there are several concerns regarding the manuscript:
1) I think that PRESTO resembles with the Mixture of Experts (MoE) family of methods. Nevertheless, the paper does not include any reference and comparison with MoE.
2) The authors compare only with the typical (trivial) case where partitioning is first obtained using clustering and then a separate model is trained on each partition. Even in this case, the results in Table 1 do not indicate any particular accuracy improvement.
4) The time complexity of the method should be specified and compared to the methods in Table 1. It should be noted that the method must run for various values of K in order for the best number of partitions to be determined through cross-validation.
5) I do not think that the method could be an efficient alternative for k-means of Bergman clustering as mentioned in section 2.3.
6) If mixture of SVMs is a special case, why not implementing PRESTO with SVMs as classifiers and compare with the original mixture of SVMs method?
7) In my opinion PRESTO does not perform mixture modeling. Mixture models are statistical models, PRESTO is a domain decomposition approach.
8) The intuition behind the a_G hyperparameter should be provided.
9) Does PRESTO algorithm converge? What is the role of the 'Iterations' parameter in Algorithm 1.

---

> ### Author Response · Authors · 2022-11-18
> **Response to Reviewer f3vu (Part-1/2)**
>
> We would like to thank the reviewer for the comments, which would improve our paper.
>
> >*Method presentation (section 3.2) and algorithm description are not clear, thus they are difficult to comprehend.*
>
> We are sorry that the algorithm appeared hard. In the following, we first elaborate on the motivation/construction of the modular upper bound and then describe the algorithm more lucidly.
>
> First, note that we are aiming to *minimize* a monotone approximate-submodular function $G$. In contrast to submodular or approximate submodular maximization where one can use greedy algorithms, *minimizing* the approximate submodular function $G$ needs a completely different approach. Here, we resort to minimizing an upper bound $m$ of $G$— reducing the value of this upper bound $m$ will ensure the value of the underlying function $G$ remains low. Now, such an upper bound $m$ has to be such that the gap between $m$ and $G$ is low and maximizing $m$ is convenient. To this aim, we choose $m$ to be modular. One can connect such a modular approximation of a set function to a simple linear approximation of a complex nonlinear function in the context of continuous optimization.
>
> $\newline$
>
> Now minimizing the modular function $m_{\hat{\mathbb{S}}}[\mathbb{S}]$ wrt $\mathbb{S}$ is equivalent to taking the minimum of [$m_{\hat{\mathbb{S}}}[(i,k)$] over different $(i,k)$ pairs ( since, $modular(Set) =  \sum_{e\in Set} modular(e)$). In each iteration of Algorithm 1, we simply compute the minimum of $m$ and update  $\mathbb{S}$ accordingly. Note that:
>
> $m^G   _{\hat{\mathbb{S}}}[\mathbb{S}]=     G (\hat{\mathbb{S}}) -  \sum _{(i,k) \in \hat{\mathbb{S}}} \alpha _G G((i,k)| \hat{\mathbb{S}} \backslash \set{(i,k)}) + \sum _{(i,k)  \in \hat{\mathbb{S}} \cap \mathbb{S}}  \alpha _G G((i,k)| \hat{\mathbb{S}} \backslash \set{(i,k)}) +  \sum _{(i,k) \in \mathbb{S} \backslash \hat{\mathbb{S}}}  \frac{G((i,k) |\emptyset)}{\alpha _G} $
>
> $\newline$
>
> Now, maximizing $m^G  _{\hat{\mathbb{S}}}[(i,k)]$ is same as maximizing  $\alpha _G G((i,k)| \hat{\mathbb{S}} \backslash \set{(i,k)})$ or $\frac{G((i,k) |\emptyset)}{\alpha _G}$ depending on whether $(i,k)\in \hat{\mathbb{S}}$ (3nd term in above Eq) or $(i,k)\not \in \hat{\mathbb{S}}$ (4th term in the above equation). We compute these two quantities in L15 and L17 of the algorithm and store the values of m at different pairs $(i,k)$ in the matrix M. Finally, we take the minimum of M to find $(i^*,k^*)$ (L19) which indicates that the partition $k^*$ should contain $i^*$. Therefore, we include $i^*$ to
>
> $\hat{S}_{k^*}$ (L20). Now, since any instance can belong to exactly one partition, we remove $i^*$ from all other partition $k \neq k^*$ (L21). Finally, we update $\hat{\mathbb{S}}$ (L22). We added the detailed explanation in Appendix G.
>
> >*The method can not directly provide the partition assignments of new examples. Therefore an additional classification model should be trained for the assignment task.*
>
> Yes, we did use an additional classification model— but that is not a weakness. The additional classifier is fairly light– in our paper, we just use a single layer linear neural network as the additional classifier. It does not provide any additional complexity.
>
> >*PRESTO is not compared to various methods belonging to the Mixture of Experts category.*
>
> We compared our method against an MoE baseline ( Outrageously Large Neural Networks: The Sparsely-Gated Mixture-of-Experts Layer : https://openreview.net/forum?id=B1ckMDqlg ) during the rebuttal period. Following are the accuracy results.
>
> ||CIFAR10|PathMNIST|DermaMNIST | SVHN
> |-|-|-|-|-|
> |PRESTO|90.04|83.70|75.61|88.11
> |MoE baseline|88.73|81.29|72.37|85.68
>
> We added the results in the Table 1.
>
> >*Time complexity is not reported.*
>
> Although PRESTO performs data partitioning while simultaneously learning a set of mixture of models,  the partitioning time  (lines 19-22 in Algorithm 1) is negligible as the bulk of the time is devoted to training (lines 10-18 in Algorithm 1). Since the training stage is common to all the methods, PRESTO does not provide much disadvantage in terms of the time. The table below shows the training time per iteration.
>
> ||CIFAR10|PathMNIST|DermaMNIST| SVHN|
> |-|-|-|-|-
> |PRESTO |9.6 min|16.6 min|1.65 min|14.6 min|
> |EqKMeans|9.4 min|16.6 min|1.49 min|14.52 min|
> |Kmeans++|7.96 min|16.6 min|1.43 min|14.46 min|
>
> We have added these results to Appendix I.

---

> > ### Author Response · Authors · 2022-11-18
> > **Response to Reviewer f3vu (Part-2/2)**
> >
> > [continued from Part-1]
> >
> > >*The results in Table 1 do not indicate any particular accuracy improvement.*
> >
> > Indeed, the performance between the second best baseline and our method is close. However, note that there is no consistent winner among the baselines across all datasets. Hence, the choice of the optimal clustering baselines differs across different datasets. In this context, our method provides optimal clustering. From a practitioner’s  viewpoint, one does not have to check which clustering algorithm to apply for partitioning. Our proposal provides a meta algorithm which can find these clusters— it can automatically find which clustering protocol one should adopt for the underlying dataset.
> >
> > The reason behind the marginal gain of our algorithm is lack of heterogeneity in the datasets. The datasets are extremely balanced and there was a lack of noise. Thus, during rebuttal times, we performed experiments where we added different amounts of label noise. Specifically, we changed each label y to a wrong label y’ with probability 10%  and then probe the performance. Results are as follows for the CIFAR10 and DermaMNIST Dataset.
> >
> > | Method | CIFAR10 | DermaMNIST |
> > | - | - | - |
> > | PRESTO       | 89.01 | 75.31
> > | EqKMeans     | 88.49 | 74.46
> > | KMeans++      |  88.08 | 74.11
> > | Agglomerative |  87.59 | 74.46
> >
> > We updated these results in Appendix H.
> >
> > >*I do not think that the method could be an efficient alternative for k-means of Bergman clustering as mentioned in section 2.3.*
> >
> > No, we are not suggesting that our method is an efficient alternative  for k-means or Bregman clustering. In section 2.3, our message was that for different loss and model architectures, our model can take the shape of different known instances. However, the framework considered in our paper is more generic— it can be used to find the optimal clustering models, be it Kmeans or Bregman clustering or mixture of SVM.
> >
> > Now, let us say Bregman clustering turns out to be the best clustering method  for a dataset. Then, of course, an algorithm specifically designed for Bregman clustering would work best. However, when we do not know the best possible choice of clustering in a supervised learning setup, our method provides a tool to find the optimal partition guided by the underlying loss and the K-model architectures.
> >
> > >*If mixture of SVMs is a special case, why not implementing PRESTO with SVMs as classifiers and compare with the original mixture of SVMs method?*
> >
> > We did not do this for real data, because support vector machines are too simple to fit the complex datasets we consider. Instead we performed experiments with synthetic dataset, where we implemented PRESTO with SVMs.  Please refer to Appendix E (Figure 6). We observe that PRESTO with SVM is able to identify the best possible partitions almost perfectly.
> >
> > >*In my opinion PRESTO does not perform mixture modeling. Mixture models are statistical models, PRESTO is a domain decomposition approach.*
> >
> > We agree that most mixture models are statistical models and PRESTO can be seen as a domain decomposition approach. However, note that in literature, there are also non-statistical models (Pal et al 2022) which have been termed as mixture models. This is why we used this as a mixture model.
> >
> > Soumyabrata Pal, Arya Mazumdar, Rajat Sen, and Avishek Ghosh. On learning mixture of linear regressions in the non-realizable setting (https://proceedings.mlr.press/v162/pal22b/pal22b.pdf)
> >
> > >*The intuition behind the a_G hyperparameter should be provided*
> >
> > Note that in submodular function $G$, the ratio between $G(T \cup e) - G(T)$ and $G(S \cup e) - G (S)$ is greater than 1. But for approximate submodular functions this ratio is always less than 1. The value $a_G$ indicates the maximum value of this ratio. If this value is high, the function becomes closer to submodular function and the approximation guarantee becomes better.
> >
> > >*Does PRESTO algorithm converge? What is the role of the 'Iterations' parameter in Algorithm 1.*
> >
> > PRESTO does converge. Please refer to the theoretical guarantee of Page no. 25 Appendix C.4 of this paper: https://arxiv.org/abs/2106.12491
> >
> > ‘Iteration’ corresponds to the number of times PRESTO is invoked to  update the partition $\mathbb{S}$.

---

> > > ### Comment · Reviewer_f3vu · 2022-12-07
> > > **repsonse to author rebuttal**
> > >
> > > I have read the author rebuttal and the revised version which improves the original version. I still have concerns about the paper such as:
> > > - comparison with MoEs: in my opinion MoEs are very close to PRESTO, however there is no mentioning in the related work section and  there is no qualititative comparison to MoEs. Why PRESTO is expected to perform better than MOEs?
> > > - If the PRESTO converges, why the number of iterations is used as a termination criterion?
> > > - I insist that mixture models are well-defined statistical models. Presto does not train a mixture model.
> > > - I still don't like the idea of training an additional model for the assignment task. Is there any guarantee that this classification problem will be an easy one to be solved by a linear classifier?

---

> > > > ### Author Response · Authors · 2022-12-07
> > > > **Response to concerns**
> > > >
> > > > Thanks for your response. We clarify your concerns as follows:
> > > >
> > > > > *comparison with MoEs: in my opinion MoEs are very close to PRESTO, however there is no mentioning in the related work section and there is no qualititative comparison to MoEs. Why PRESTO is expected to perform better than MOEs?*
> > > >
> > > > We initially thought that the reviewer only requested for experimental comparison against MoE models which is why we did not add a discussion in the related work.  We now better understand the reviewer’s concern.
> > > >
> > > > We describe the qualitative differences between MoE model and our proposal as follows:
> > > >
> > > > Let us consider the following MoE model, we have $y$ is computed using $ \sum _{k=1} ^M G(x)_k H(x) _k  $. Here, $M$ is the number of experts, $G$ is a gating network which is typically modelled as a probability distribution over the experts and  $H$ is the $k$-th expert model.  Here, the predicted probability for $y$ is computed by averaging over the gating probabilities. Thus, the corresponding training loss  becomes $\sum _{(x,y) \in D}\ell (\sum _{k=1} ^M G(x) _kH(x) _k, y)$. This can also be written as $\sum _{(x,y) \in D} \ell (E _{k \sim G(x)} [H(x) _k], y)$ (Here $E$ is expectation).
> > > >
> > > > Now, instead of  computing average over the expert networks and then computing loss, we can first compute loss for a given expert and then compute average over experts as follows: $\sum _{(x,y) \in D} E _{k \sim G(x)} [\ell ( H(x) _k, y)]$.  By Jensen inequality, $\sum _{(x,y) \in D} E _{k \sim G(x)} [\ell ( H(x) _k, y)]$ > $\sum _{(x,y) \in D} \ell (E _{k \sim G(x)} [H(x) _k], y)$) and therefore, $\sum _{(x,y) \in D} E _{k \sim G(x)} [\ell ( H(x) _k, y)]$ **minimises the error more aggressively**.
> > > >
> > > > In our setup, we minimize the second loss $\sum _{(x,y) \in D} E _{k \sim G(x)} [\ell ( H(x) _k, y)]$ in the combinatorial sense. Here, we do not use a neural gating network. Instead, for each x, we assume $p _x(k)$ to be the probability that the instance “x” belongs to expert $k$. Here, $p _x$ and $p _{x'} $ do not share any parameter— they are different variables. Note that, the continuous optimization problem
> > > >
> > > > $\min  _{G,H} \sum _{(x,y) \in D} E _{k \sim G(x)} [\ell ( H(x) _k, y)]$--- (1)
> > > >
> > > > can be seen as an approximation of the combinatorial optimization problem given as follows:
> > > >
> > > >  $\min _{[p _x] _{x\in D} ,H} \sum _{(x,y) \in D} \sum _{k=1} ^M p _x[k][\ell ( H(x) _k, y)]$ subject to $\sum _{k=1} ^M p _x(k) =1$ and $p _x (k) \ge 0 $ ---(2)
> > > >
> > > > (2) is an approximation of (1) because the gating network $G$ is parameterized to capture the variables $p _x$ via a shared neural network.  Since (2) is a linear program wrt $p _x$, the solution of $p _x$ lies in the vertices of the linear constraints and thus at optimal point, we have $p _x(k) =1$ or $0$. This directly reduces our problem to the following optimization:
> > > >
> > > >
> > > >  $\min  _{S_k ,H} \sum _{(x,y) \in S_k}  \ell ( H(x) _k, y) $
> > > >
> > > > where $S_k = ( (x,y): p _x(k) = 1 )$
> > > >
> > > > Thus, Presto offers competitive advantage over MoE models because (1) it minimizes a more aggressive loss which is the average of losses on different subsets; and (2) the gating network is rather a soft version of our proposed combinatorial instance subset assignment problem.
> > > >
> > > > > *If the PRESTO converges, why the number of iterations is used as a termination criterion?*
> > > >
> > > > In principle, yes we should use convergence criteria: |loss_at_iteration_r - loss_at_iteration_{r+1}| < epsilon*loss_at_iteration_r as the stopping criteria. However, in practice, we found by iteration = 30, we reach epsilon < 1e-3 for all datasets. This is why, we used iterations as the stopping criteria. In the revised version, we can clarify this.
> > > >
> > > > > *I insist that mixture models are well-defined statistical models. Presto does not train a mixture model.*
> > > >
> > > > Frankly, we don’t have a strong opinion here. To the best of our understanding, similar setup was also mentioned as mixture model in the paper by Pal et al. [a]. This is why we call our setup a mixture model.
> > > > https://arxiv.org/abs/2205.13166
> > > >
> > > > If the reviewer really insists, we can write Presto as a domain decomposition approach in the final version of our paper, if accepted.
> > > >
> > > > > *I still don't like the idea of training an additional model for the assignment task. Is there any guarantee that this classification problem will be an easy one to be solved by a linear classifier?*
> > > >
> > > > In all datasets (real and synthetic), we find that linear classifiers provide very good accuracy.  Even upon adding label noises during experiments at the rebuttal time, we find linear classifiers provide good accuracy. We initially tried with nearest neighbor classifiers, however, they did not work well for real data because of high dimensionality.

---

> > > > > ### Comment · Reviewer_f3vu · 2022-12-09
> > > > > **response to authors**
> > > > >
> > > > > Thanks for the reply. The revised paper has been improved over the initial version. However, experimental results still do not indicate sufficient superiority over the simple two-stage (first cluster, then learn) decomposition approach. The fact that the assignment problem is solved by a linear classifier indicates that the obtained partitions are simple (linear), therefore such partitions could also be obtained by a typical clustering method.

---

### Official Review · Reviewer_dXtq · 2022-10-27

**Confidence:** 4
**Correctness:** 4
**Technical Novelty And Significance:** 3
**Empirical Novelty And Significance:** 3
**Recommendation:** 6

**Clarity, Quality, Novelty And Reproducibility:**

Clarity: The submission is clearly written and well-organized. I do have one qualm with the paper though. The algorithm discussed in the paper builds heavily on the some prior algorithms (Iyer et al, 2013 a;b, Durga et al. (2021)) but there is no intuition provided in the paper for either the majorization-minimization approach or the modular upper bound. It would be nice if you could add a paragraph or two briefly describing these things (appendix is also fine). I was not familiar with any of these works and it made it hard to follow the algorithm.

Quality: The submission is technically sound. All claims are well-supported with proofs and detailed experiments.

Novelty: I am not an expert in this area so I am not entirely sure about other related work. I have a question here, which I hope the authors can answer. Was the specific problem discussed in this paper studied by any other work? To additionally clarify, I was curious if even this baseline approach with $k$-means++ and learning a model for each partition was discussed in any prior work.

Reproducibility: Proofs are provided and also the code for experiments is available.

Typos/minor issues:

Page 2: “key rational” -> “key rationale”

Duplicate citations for the same paper:

Soumyabrata Pal, Arya Mazumdar, Rajat Sen, and Avishek Ghosh. On learning mixture of linear regressions in the non-realizable setting. In International Conference on Machine Learning, pp. 17202–17220. PMLR, 2022a.
Soumyabrata Pal, Arya Mazumdar, Rajat Sen, and Avishek Ghosh.

On learning mixture of linear regressions in the non-realizable setting. In Kamalika Chaudhuri, Stefanie Jegelka, Le Song, Csaba Szepesvari, Gang Niu, and Sivan Sabato (eds.), Proceedings of the 39th International Conference on Machine Learning, volume 162 of Proceedings of Machine Learning Research, pp. 17202–17220. PMLR, 17–23 Jul 2022b. URL https://proceedings.mlr.press/ v162/pal22b.html.

Page 4, Section 2.3, K-Means Clustering: “turns out that $\theta_k = \mu_k$, and the cluster means and $\ell$…” -> “turns out that $\theta_k = \mu_k$, the cluster means, and $\ell$…”

Section 2.3 has repeated paragraph titles “Mixtures of SVMs and linear regressions” and “Mixture of Linear Regression”

Definition 1 on Page 5: “f is $\alpha$-submodular submodular”

Appendix A.1 (1) “..., then $T \subset I$” -> “..., then $S \in I$”
                        (2) “ If $S \subset I$ …” -> “If $S \in I$”

Theorem 2. $\underline{\ell}_{min}$ definition is a bit unclear. The underscript in min should be expanded and there should be a ] after $||\theta_k||^2$

(3) of Theorem 2: $\kappa_G(S) \leq \kappa_G^*$. Here, should $S$ be $\mathbb{S}$?

Just before equation (7): “optimal solution $\mathbb{S}^*$ Thus, the optimization (5) becomes” : Period missing between ‘$\mathbb{S}^*$’ and ‘Thus’

Algorithm 1 Line 2

Missing period at the end of Theorem 4.


**Strength And Weaknesses:**

Strengths: Interesting paper which is overall well-written and studies a significant problem. Such algorithms can be useful when there is a large dataset, which was potentially generated from heterogeneous data-sources.

Weaknesses:
Although the problem studied by the paper is interesting, the algorithm they provide (PRESTO) is a bit complicated. Some of the simpler baselines (like the k-means++ based approach) give performance which is pretty close to their approach. I am very skeptical if any practitioner would bother using their approach for the marginal improvement in utility compared to the simpler baselines. Also, can the authors comment on the time taken by their algorithm when compared to their baselines?

Although the paper provides a approximation bound (Theorem 4), I am skeptical if this is of any use in practice. More concretely, for some of the loss functions and the data-sets which the authors consider in the experiments, what is the value of this approximation factor?



**Summary Of The Paper:**

This paper studies the following problem: Given a labeled dataset, a parameter $K$, and $K$ model architectures, we want to partition the data into $K$ partitions and also learn all the parameters for each model corresponding to the partitions. To help understand the problem and compare their approach, I will discuss a simple baseline they use in their paper.

Given a labeled dataset, first run $k$-means++ using only the features (ignoring the labels) to form the $K$ partitions, then independently train a model for each of the partitions (now also using the labels). Note that in this approach, we have completely ignored the labels in the data-partitioning step. What this paper does is to instead also incorporate the labels and simultaneously learn the model parameters and the data-partitions.

**Summary Of The Review:**

Overall interesting paper. I am currently leaning to accept but would be willing to increase my score after the author response.

---

> ### Author Response · Authors · 2022-11-18
> **Response to Reviewer dXtq (Part-1/2)**
>
> We would like to thank the reviewer for the comments/suggestions, which would indeed improve our paper.
>
> >*Although the problem studied by the paper is interesting, the algorithm they provide (PRESTO) is a bit complicated…
> I do have one qualm with the paper though. The algorithm discussed in the paper builds heavily on the some prior algorithms (Iyer et al, 2013 a;b, Durga et al. (2021)) but there is no intuition provided in the paper for either the majorization-minimization approach or the modular upper bound. It would be nice if you could add a paragraph or two briefly describing these things (appendix is also fine). I was not familiar with any of these works and it made it hard to follow the algorithm.*
>
> We are sorry that the algorithm appeared hard. In the following, we first elaborate on the motivation/construction of the modular upper bound and then describe the algorithm more lucidly.
>
> First, note that we are aiming to *minimize* a monotone approximate-submodular function $G$. In contrast to submodular or approximate submodular maximization where one can use greedy algorithms, *minimizing* the approximate submodular function $G$ needs a completely different approach. Here, we resort to minimizing an upper bound $m$ of $G$— reducing the value of this upper bound $m$ will ensure the value of the underlying function $G$ remains low. Now, such an upper bound $m$ has to be such that the gap between $m$ and $G$ is low and maximizing $m$ is convenient. To this aim, we choose $m$ to be modular. One can connect such a modular approximation of a set function to a simple linear approximation of a complex nonlinear function in the context of continuous optimization.
>
> $\newline$
>
> Now minimizing the modular function $m_{\hat{\mathbb{S}}}[\mathbb{S}]$ wrt $\mathbb{S}$ is equivalent to taking the minimum of [$m_{\hat{\mathbb{S}}}[(i,k)$] over different $(i,k)$ pairs ( since, $modular(Set) =  \sum_{e\in Set} modular(e)$). In each iteration of Algorithm 1, we simply compute the minimum of $m$ and update  $\mathbb{S}$ accordingly. Note that:
>
> $m^G   _{\hat{\mathbb{S}}}[\mathbb{S}]=     G (\hat{\mathbb{S}}) -  \sum _{(i,k) \in \hat{\mathbb{S}}} \alpha _G G((i,k)| \hat{\mathbb{S}} \backslash \set{(i,k)}) + \sum _{(i,k)  \in \hat{\mathbb{S}} \cap \mathbb{S}}  \alpha _G G((i,k)| \hat{\mathbb{S}} \backslash \set{(i,k)}) +  \sum _{(i,k) \in \mathbb{S} \backslash \hat{\mathbb{S}}}  \frac{G((i,k) |\emptyset)}{\alpha _G} $
>
> $\newline$
>
> Now, maximizing $m^G  _{\hat{\mathbb{S}}}[(i,k)]$ is same as maximizing  $\alpha _G G((i,k)| \hat{\mathbb{S}} \backslash \set{(i,k)})$ or $\frac{G((i,k) |\emptyset)}{\alpha _G}$ depending on whether $(i,k)\in \hat{\mathbb{S}}$ (3nd term in above Eq) or $(i,k)\not \in \hat{\mathbb{S}}$ (4th term in the above equation). We compute these two quantities in L15 and L17 of the algorithm and store the values of m at different pairs $(i,k)$ in the matrix M. Finally, we take the minimum of M to find $(i^*,k^*)$ (L19) which indicates that the partition $k^*$ should contain $i^*$. Therefore, we include $i^*$ to $\hat{S} _{k^*}$ (L20 in Algorithm1). Now, since any instance can belong to exactly one partition, we remove $i^*$ from all other partition $k \neq k^*$ (L21). Finally, we update $\hat{\mathbb{S}}$ (L22). We added the detailed explanation in Appendix G.
>
> >*Some of the simpler baselines (like the k-means++ based approach) give performance which is pretty close to their approach. I am very skeptical if any practitioner would bother using their approach for the marginal improvement in utility compared to the simpler baselines. Also, can the authors comment on the time taken by their algorithm when compared to their baselines?*
>
> Indeed, the performance between the second best baseline and our method is close. However, note that there is no consistent winner among the baselines across all datasets. Hence, the choice of the optimal clustering baselines differs across different datasets. In this context, our method provides optimal clustering. From a practitioner’s  viewpoint, one does not have to check which clustering algorithm to apply for partitioning. Our proposal provides a meta algorithm which can find these clusters— it can automatically find which clustering protocol one should adopt for the underlying dataset.
>
> The reason behind the marginal gain of our algorithm is lack of heterogeneity in the datasets. The datasets are extremely balanced and there was a lack of noise. Thus, during rebuttal times, we performed experiments where we added different amounts of label noise. Specifically, we changed each label y to a wrong label y’ with probability 10%  and then probe the performance. Results are as follows for the CIFAR10 and DermaMNIST Dataset.
>
> | Method | CIFAR10 | DermaMNIST |
> | - | - | - |
> | PRESTO       | 89.01 | 75.31
> | EqKMeans     | 88.49 | 74.46
> | KMeans++      |  88.08 | 74.11
> | Agglomerative |  87.59 | 74.46
>
> We updated these results in Appendix H.

---

> > ### Author Response · Authors · 2022-11-18
> > **Response to Reviewer dXtq (Part-2/2)**
> >
> > [continued from Part-1]
> >
> > >*Was the specific problem discussed in this paper studied by any other work? To additionally clarify, I was curious if even this baseline approach with  k-means++ and learning a model for each partition was discussed in any prior work.*
> >
> > Our problem considers learning a mixture model with general loss and model architecture on a non-realizable setting, where we do not assume any generative distribution of the data. To the best of our knowledge, only Pal et al (2022a) considered a similar non-realizable setting, however the paper focuses only on the linear regression task. PRESTO on the other hand is
> > designed for generic deep learning problems using even non-linear architectures.
> >
> > To the best of our knowledge, we did not come across any work that partitioned the dataset and then learned a model for each partition (like kmeans++).
> >
> > >*Also, can the authors comment on the time taken by their algorithm when compared to their baselines?*
> >
> > Although PRESTO performs data partitioning while simultaneously learning a set of mixture of models,  the partitioning time  (lines 19-22 in Algorithm 1) is negligible as the bulk of the time is devoted to training (lines 10-18 in Algorithm 1). Since the training stage is common to all the methods, PRESTO does not provide much disadvantage in terms of the time. The table below shows the training time per iteration.
> >
> > ||CIFAR10|PathMNIST|DermaMNIST| SVHN|
> > |-|-|-|-|-
> > |PRESTO |9.6 min|16.6 min|1.65 min|14.6 min|
> > |EqKMeans|9.4 min|16.6 min|1.49 min|14.52 min|
> > |Kmeans++|7.96 min|16.6 min|1.43 min|14.46 min|
> >
> > We have added these results to Appendix I.
> >
> > >*Although the paper provides an approximation bound (Theorem 4), I am skeptical if this is of any use in practice. More concretely, for some of the loss functions and the data-sets which the authors consider in the experiments, what is the value of this approximation factor?*
> >
> > We found that the values of the approximation bounds are quite reasonable.  For CIFAR10 the value is 1.73;  for PathMNIST the value is 2.03; for DermaMNIST the value is 1.79 and for SVHN the value is 2.38.
> >
> > $\newline$
> >
> > >*Typo and Minor Issues*
> >
> > Minor issues and typos suggested by the reviewer have been addressed and rectified in the revised version.

---

> > > ### Comment · Reviewer_dXtq · 2022-12-09
> > > **Acknowledgment of response**
> > >
> > > I wanted to acknowledge that I read the author's response and also the reviews by other reviewers. It is good to know that the time taken by the algorithm is close to the simpler baselines. I thought that it might take much longer...
> > >
> > > One of my key concerns regarding the paper still seems to be unaddressed after the response, which is
> > >
> > > > Some of the simpler baselines (like the k-means++ based approach) give performance which is pretty close to their approach. I am very skeptical if any practitioner would bother using their approach for the marginal improvement in utility compared to the simpler baselines.
> > >
> > > Unfortunately, the new results with added noise haven't convinced me otherwise.

---

### Author Response · Authors · 2022-11-29
**Authors-to-reviewers: Did our responses address all your concerns?**

Dear Reviewers,

Thank you for all your suggestions and comments. We attempted to address them with due diligence. Could you please acknowledge our rebuttal and let us know whether our responses addressed all your concerns? We would be more than happy to respond if you have further questions/concerns.

Thanks,
Authors

---

### Decision · Program_Chairs · 2023-01-20

**Decision:**

Reject

**Justification For Why Not Higher Score:**

The empirical results are not very compelling. The algorithm is quite complicated and it achieves only modest improvements over much simpler baselines. The theoretical results do not tell a clean story, since the assumptions are quite strong and the approximation guarantees are quite large.

**Justification For Why Not Lower Score:**

N/A

**Metareview: Summary, Strengths And Weaknesses:**

This paper introduces a novel data partitioning problem, where the goal is to partition a dataset into k groups so as to minimize a joint objective function that measures the loss of the best model on each group, and has a regularization term. This framework nominally contains many other well-studied problems, like clustering, mixtures of support vector machines and mixtures of linear regression. The main theoretical contribution is an algorithm which they show, for large enough values of the regularizing coefficient, achieves a bounded approximation ratio. This is achieved by connecting it to submodular optimization with bounded curvature subject to a matroid constraint. However the reviewers found the empirical results to be unconvincing.

**Summary Of Ac-Reviewer Meeting:**

The reviewers mostly focused on the empirical aspects, and no one was willing to champion the paper. I read through the theoretical contributions and did not find these compelling either, mostly because the approximation factor is quite large even under rather strong assumptions and it is not clear if it yields any meaningful guarantees.